# Beware of the Simulated DAG!
# Causal Discovery Benchmarks May Be Easy To Game

**Alexander G. Reisach[1,2]**         **Christof Seiler[2,3]**         **Sebastian Weichwald[1]**

[1]Department of Mathematical Sciences, University of Copenhagen, Denmark
[2]Department of Data Science and Knowledge Engineering, Maastricht University, The Netherlands
[3]Mathematics Centre Maastricht, Maastricht University, The Netherlands

## Abstract

Simulated DAG models may exhibit properties that, perhaps inadvertently, render their structure identifiable and unexpectedly affect structure learning algorithms. Here, we show that marginal variance tends to increase along the causal order for generically sampled additive noise models. We introduce *varsortability* as a measure of the agreement between the order of increasing marginal variance and the causal order. For commonly sampled graphs and model parameters, we show that the remarkable performance of some continuous structure learning algorithms can be explained by high varsortability and matched by a simple baseline method. Yet, this performance may not transfer to real-world data where varsortability may be moderate or dependent on the choice of measurement scales. On standardized data, the same algorithms fail to identify the ground-truth DAG or its Markov equivalence class. While standardization removes the pattern in marginal variance, we show that data generating processes that incur high varsortability also leave a distinct covariance pattern that may be exploited even after standardization. Our findings challenge the significance of generic benchmarks with independently drawn parameters. The code is available at https://github.com/Scriddie/Varsortability.

## 1 Introduction

**Causal structure learning** aims to infer a causal model from data. Academic disciplines anywhere from biology, medicine, finance, to machine learning are interested in causal models [Rothman et al., 2008, Imbens and Rubin, 2015, Sanford and Moosa, 2012, Schölkopf, 2019]. Causal models not only describe the observational joint distribution of variables but also formalize predictions under interventions and counterfactuals [Spirtes et al., 2000, Pearl, 2009, Peters et al., 2017]. Directed acyclic graphs (DAGs) are common to represent causal structure: nodes represent variables and directed edges point from cause to effect representing the causal relationships. This graphical representation rests on assumptions which have been critically questioned, for example by Dawid [2010]. Inferring causal structure from observational data is difficult: Often we can only identify the DAG up to its Markov equivalence class (MEC) and finding high-scoring DAGs is NP-hard [Chickering, 1996, Chickering et al., 2004]. Here, we focus on learning the DAG of linear additive noise models (ANM).

**Data scale and marginal variance** may carry information about the data generating process. This information can dominate benchmarking results, such as, for example, the outcome of the NeurIPS Causality 4 Climate competition [Runge et al., 2020]. Here, the magnitude of regression coefficients was informative about the existence of causal links such that ordinary regression-based methods on raw data outperformed causal discovery algorithms [Weichwald et al., 2020]. Multiple prior works

state the importance of data scale for structure learning either implicitly or explicitly. Structure identification by *ICA-LiNGAM* [Shimizu et al., 2006], for example, is susceptible to rescaling of the variables. This motivated the development of *DirectLiNGAM* [Shimizu et al., 2011], a scale-invariant causal discovery algorithm for linear non-Gaussian models. The causal structure of ANMs is proven to be identifiable given the noise scale (cf. Section 2.2). Yet, such identifiability results require knowledge about the ground-truth data scale.

**Simulated DAGs** may be identifiable from marginal variances under generic parameter distributions. An instructive example is the causal graph $A \rightarrow B$ with structural equations $A = N_A$ and $B = wA + N_B$ with $w \neq 0$ and independent zero-centered noise variables $N_A, N_B$. The mean squared error (MSE) of a model $X \rightarrow Y$ is given by $\mathrm{MSE}\,(X \rightarrow Y) = \mathrm{Var}(X) + \mathrm{Var}(Y|X)$. It holds that $\mathrm{MSE}\,(A \rightarrow B) < \mathrm{MSE}\,(B \rightarrow A) \iff \mathrm{Var}(A) < \mathrm{Var}(B) \iff (1-w^2)\,\mathrm{Var}(N_A) < \mathrm{Var}(N_B)$ (see Appendix A). Deciding the directionality of the edge between $A$ and $B$ based on the MSE amounts to inferring an edge from the lower-variance variable to the higher-variance variable. For error variances $\mathrm{Var}(N_A) \leq \mathrm{Var}(N_B)$ and any non-zero edge weight $w$, the MSE-based inference is correct. This resembles known scale-based identifiability results based on equal or monotonically increasing error variances [Peters and Bühlmann, 2014, Park, 2020]. However, if the observations of $A$ were multiplied by a sufficiently large constant, the MSE-based inference would wrongly conclude that $A \leftarrow B$. This is problematic since simply choosing our units of measurement differently may change the scale and variance of $A$. Arguably, this is often the case for observations from real-world systems: There is no canonical choice as to whether we should pick meters or yards for distances, gram or kilogram for weights, or yuan or dollar as currency. A researcher cannot rely on obtaining the same results for different measurement scales or after re-scaling the data when applying any method that leverages the data scale (examples include Peters and Bühlmann [2014], Park [2020], or Zheng et al. [2018], who employ the least squares loss studied by Loh and Bühlmann [2014]).

**Continuous causal structure learning algorithms** optimize model fit under a differentiable acyclicity constraint [Zheng et al., 2018]. This allows for the use of continuous optimization and avoids the explicit combinatorial traversal of possible causal structures. This idea has found numerous applications and extensions [Lachapelle et al., 2019, Lee et al., 2019, Ng et al., 2020, Yu et al., 2019, Brouillard et al., 2020, Pamfil et al., 2020, Wei et al., 2020, Zheng et al., 2020, Bhattacharya et al., 2021]; Vowels et al. [2021] provide a review. *NOTEARS* [Zheng et al., 2018] uses the MSE with reference to Loh and Bühlmann [2014], while *GOLEM* [Ng et al., 2020] assesses model fit by the penalized likelihood assuming a jointly Gaussian model. On simulated data and across noise distributions, both methods recover graphs that are remarkably close to the ground-truth causal graph in structural intervention distance (SID) and structural hamming distance (SHD). We agree with the original authors that these empirical findings, especially under model misspecification and given the non-convex loss landscape, may seem surprising at first. Here, we investigate the performance under data standardization and explain how the causal order is (partially) identifiable from the raw data scale alone in common generically simulated benchmarking data.

**Contribution.** We show that causal structure drives the marginal variances of nodes in an ANM and can lead to (partial) identifiability. The pattern in marginal variances is dominant in ANM benchmark simulations with edge coefficients drawn identically and independently. We introduce varsortability as a measure of the information the data scale carries about the causal structure. We argue that high varsortability affects the optimization procedures of continuous structure learning algorithms. Our experiments demonstrate that varsortability dominates the optimization and helps achieve state-of-the-art performance provided the ground-truth data scale. Data standardization or an unknown data scale remove this information and the same algorithms fail to recover the ground-truth DAG. Even methods using a score-equivalent likelihood criterion (*GOLEM*) recover neither ground-truth DAG nor its MEC on standardized data. To illustrate that recent benchmark results depend heavily on high varsortability, we provide a simple baseline method that exploits increasing marginal variances to achieve state-of-the-art results on these benchmarks. We thereby provide an explanation for the unexpected performance of recent continuous structure learning algorithms in identifying the true DAG. Neither algorithm dominates on raw or standardized observations of the analyzed real-world data. We show how, even if data is standardized and even in non-linear ANMs, a causal discovery benchmark may be gamed due to covariance patterns. Consequently, recent benchmark results may not transfer to (real-world) settings where the correct data scale is unknown or where edge weights are not drawn independent and identically distributed (iid). We conclude that structure learning

benchmarks on ANMs with generically sampled parameters may be distorted due to unexpected and perhaps unintended regularity patterns in the data.

## 2 Background

### 2.1 Model Class

We consider acyclic linear additive noise models. Single observations are denoted by $x^{(i)} \in \mathbb{R}^d$ where $x_j^{(i)}$ denotes the $j^{\text{th}}$ dimension of the $i^{\text{th}}$ iid observation of random vector $X = [X_1, ..., X_d]^\top$. All observations are stacked as $\mathbf{X} = [x^{(1)}, ..., x^{(n)}]^\top \in \mathbb{R}^{n \times d}$ and $x_j \in \mathbb{R}^n$ refers to the $j^{\text{th}}$ column of $\mathbf{X}$. Analogously, $n^{(i)}$ denotes the corresponding $i^{\text{th}}$ iid observation of the random noise variable $N = [N_1, ..., N_d]^\top$ with independent zero-centred components. The linear effect of variable $X_k$ on $X_j$ is denoted by $w_{k \to j} = w_{kj}$. The causal structure corresponding to the adjacency matrix $W = [w_{kj}]_{k,j=1,...,d}$ with columns $w_j = [w_{k \to j}]_{k=1,...,d} \in \mathbb{R}^d$ can be represented by a directed acyclic graph $G = (V_G, E_G)$ with vertices $V_G = \{1, ..., d\}$ and edges $E_G = \{(k,j) : w_{k \to j} \neq 0\}$. Edges can be represented by an adjacency matrix $E$ such that the $(k,j)^{\text{th}}$ entry of $E^l$ is non-zero if and only if a directed path of length $l$ from $k$ to $j$ exists in $G$. For a given graph, the parents of $j$ are denoted by PA $(j)$. The structural causal model is $X = W^\top X + N$.

### 2.2 Identifiability of Additive Noise Models

Identifiability of the causal structure or its MEC requires causal assumptions. Under causal faithfulness and Markov assumptions, the causal graph can be recovered up to its MEC [Chickering, 1995, Spirtes et al., 2000]. Faithfulness, however, is untestable [Zhang and Spirtes, 2008]. Shimizu et al. [2006] show that under the assumptions of no unobserved confounders, faithfulness, linearity, and non-Gaussian additive noise, the causal graph can be recovered from data. Hoyer et al. [2009] show that this holds for any noise distribution under the assumption of strict non-linearity. This finding is generalized to post-nonlinear functions by Zhang and Hyvarinen [2009]. Peters and Bühlmann [2014] prove that the causal structure of a linear causal model with Gaussian noise is identifiable if the error variances are equal or known. Any unknown re-scaling of the data breaks this condition. For the case of linear structural causal models, Loh and Bühlmann [2014] provide a framework for DAG estimation based on a noise variance-weighted least squares score function. For ANMs, they give conditions under which the general Gaussian case can be identified via approximating it by the equal noise-variance case given knowledge of the (approximate) noise scale. Finally, subsuming further prior results on (linear) ANMs [Hoyer et al., 2009, Ghoshal and Honorio, 2017, 2018, Chen et al., 2019], Park [2020] shows that the causal structure is identifiable under regularity conditions on the conditional variances along the causal order. In particular, identifiability holds if the error variances of nodes are weakly monotonically increasing along the causal order.

### 2.3 Structure Learning Algorithms

**Combinatorial structure learning algorithms (such as *PC, FGES, DirectLiNGAM*)** separately solve the combinatorial problem of searching over structures and finding the optimal parameters for each structure. To remain computationally feasible, the search space of potential structures is often restricted or traversed according to a heuristic. One can, for example, carefully choose which conditional independence statements to evaluate in constraint-based algorithms, or employ greedy (equivalence class) search in score-based algorithms. In our experiments, we consider *PC* [Spirtes and Glymour, 1991], *FGES* [Meek, 1997, Chickering, 2002b], *DirectLiNGAM* [Shimizu et al., 2011], and a greedy DAG search (GDS) algorithm *MSE-GDS* that greedily includes those edges that reduce the MSE the most. For details see Appendix D.

**Continuous structure learning algorithms (such as *NOTEARS* and *GOLEM*)** employ continuous optimization to simultaneously optimize over structures and parameters. As a first step towards expressing causal structure learning as a continuous optimization problem, Aragam and Zhou [2015] propose $l^1$-regularization instead of the conventional $l^0$-penalty for model selection. Zheng et al. [2018] propose a differentiable acyclicity constraint, allowing for end-to-end optimization of score functions over graph adjacency matrices. We examine and compare the continuous structure learning algorithms *NOTEARS* [Zheng et al., 2018] and *GOLEM* [Ng et al., 2020]. For details see Appendix D.

*NOTEARS* [Zheng et al., 2018] minimizes the MSE between observations and model predictions subject to a hard acyclicity constraint. The MSE with respect to $W$ on observations $\mathbf{X}$ is defined as $\mathrm{MSE}_{\mathbf{X}}(W) = \frac{1}{n}\|\mathbf{X} - \mathbf{X}W\|_2^2$ where $\|\cdot\|_2 = \|\cdot\|_F$ denotes the Frobenius norm.

*GOLEM* [Ng et al., 2020] performs maximum likelihood estimation (MLE) under the assumption of a Gaussian distribution with equal (EV) or non-equal (NV) noise variances. There are soft acyclicity and sparsity constraints. The unnormalized negative likelihood-parts of the objective function are $\mathcal{L}_{EV}(W, \mathbf{X}) = \log(\mathrm{MSE}_{\mathbf{X}}(W))$ and $\mathcal{L}_{NV}(W, \mathbf{X}) = \sum_{j=1}^{d} \log\left(\frac{1}{n}\|x_j - \mathbf{X}w_j\|_2^2\right)$, respectively, omitting a $-\log(|\det(I - W)|)$ term that vanishes when $W$ represents a DAG [Ng et al., 2020].

To ease notation, we sometimes drop the explicit reference to $\mathbf{X}$ when referring to MSE, $\mathcal{L}_{EV}, \mathcal{L}_{NV}$.

## 3 Varsortability

The data generating process may leave information about the causal order in the data scale. We introduce varsortability as a measure of such information. When varsortability is maximal, the causal order is identifiable. Varsortability is high in common simulation schemes used for benchmarking causal structure learning algorithms. We describe how continuous structure learning algorithms are affected by marginal variances and how they may leverage high varsortability. This elucidates the results of continuous methods reported by Zheng et al. [2018], Ng et al. [2020], and others on raw data and predicts impaired performance on standardized data as confirmed in Section 4. We introduce *sortnregress* as simple baseline method that sorts variables by marginal variance followed by parent selection. The performance of *sortnregress* reflects the degree of varsortability in a given setting and establishes a reference baseline to benchmark structure learning algorithms against.

### 3.1 Definition of Varsortability

We propose varsortability as a measure of agreement between the order of increasing marginal variance and the causal order. For any causal model over variables $\{X_1, ..., X_d\}$ with (non-degenerate) DAG adjacency matrix $E$ we define varsortability as the fraction of directed paths that start from a node with strictly lower variance than the node they end in, that is,

$$v := \frac{\sum_{k=1}^{d-1} \sum_{i \to j \in E^k} \mathrm{increasing}(\mathrm{Var}(X_i), \mathrm{Var}(X_j))}{\sum_{k=1}^{d-1} \sum_{i \to j \in E^k} 1} \in [0, 1] \text{ where } \mathrm{increasing}(a, b) = \begin{cases} 1 & a < b \\ 1/2 & a = b \\ 0 & a > b \end{cases}$$

For example, we calculate the varsortability as $v = \frac{1+1+1}{1+1+1+1} = \frac{3}{4}$ given the causal graph below.

Varsortability equals one if the marginal variance of each node is strictly greater than that of its causal ancestors. Varsortability equals zero if the marginal variance of each node is strictly greater than that of its descendants. Varsortability does not depend on choosing one of the possibly multiple causal orders and captures the overall agreement between the partial order induced by the marginal variances and all pathwise descendant relations implied by the causal structure. In the two-node introductory example (cf. Section 1), varsortability $v = 1$ is equivalent to

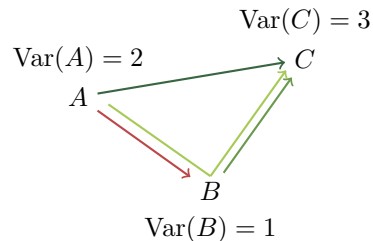

$$\mathrm{Var}(A) < \mathrm{Var}(B) \iff \mathrm{Var}(N_A) < w^2 \mathrm{Var}(N_A) + \mathrm{Var}(N_B)$$

where $A$ and $B$ are nodes in the causal graph $A \xrightarrow{w} B$ with noise variances $\mathrm{Var}(N_A)$ and $\mathrm{Var}(N_B)$.

We can also understand varsortability as a property of the distribution of graphs and parameters that we sample from for benchmarks on synthetic data. The distribution of weights and noise variances determines whether the causal order of any two connected nodes in the graph agrees with the order of increasing marginal variance and in turn determines the varsortability of the simulated causal models. We observe that even for modest probabilities of any two neighboring nodes being correctly ordered by their marginal variances, the variance of connected nodes tends to increase quickly along the causal order for many ANM instantiations (cf. Appendix G.3). For a heuristic explanation, recall that we obtain the marginal variance of a node by adding the variance contribution of all its ancestors to the

node's own noise variance; to obtain the variance contribution of an ancestor, we take the product of the edge weights along each directed path from ancestor to node, sum these path coefficient products, square, and multiply with the ancestor's noise variance. While the sum of path coefficient products may vanish or be small such that the variance contribution of an ancestor cancels out or is damped across the different connecting paths, we find it is unlikely if edge weights are drawn independently (cf. Meek [1995] for why exact cancellation and faithfulness violations are unlikely). Furthermore, the further apart a connected pair of nodes, the more variance may be accumulated in the descendant node along all incoming paths in addition to one ancestor's (possibly damped) variance contribution further fueling the tendency for descendant nodes to have higher variance than their ancestors. In practice we indeed find that an ordering of nodes by increasing marginal variance closely aligns with the causal order for commonly simulated linear and non-linear ANMs (cf. Appendix G).

## 3.2 Varsortability and Identifiability

If varsortability $v = 1$, the causal structure is identifiable. It can be recovered by ordering the nodes by increasing marginal variance and regressing each node onto its predecessors using conventional sparse regression approaches. The causal structure learning problem is commonly decoupled into causal order estimation and parent selection [Shimizu et al., 2006, Shojaie and Michailidis, 2010, Bühlmann et al., 2014, Chen et al., 2019, Park, 2020]. This decoupling is further warranted, since we only need the causal ordering to consistently estimate interventional distributions [Bühlmann et al., 2014, Section 2.6]. At $v = 1$, an ordering by marginal variance is a valid causal ordering. Given a causal ordering, one can construct a fully connected DAG and use parent selection to prune edges and reconstruct the graph in the sample limit under mild assumptions. Bühlmann et al. [2014, Section 2.5] discuss parent selection and Shojaie and Michailidis [2010] establish the consistency of an adaptive lasso approach for edge selection given a valid causal order. The identifiability conditions by Park [2020] are closely related to varsortability, though not equivalent as we prove in Appendix C. Identifiability of the causal order is immediate if varsortability $v = 1$, though, this shares severe drawbacks with other identifiability conditions that rely on data scale by Peters and Bühlmann [2014], Loh and Bühlmann [2014], and Park [2020]. First, it is difficult to verify or assess the plausibility of assumptions about the correctness or suitability of the data scale for any given dataset. Second, any unknown rescaling may break previously met identifiability conditions. Third, even if variables are on the same measurement scale the units may not correctly capture the ground-truth causal scale. For example, a dartist's distance from the dartboard may affect the precision of their throw measured by the distance between hit and target. Here, the effect variable's marginal variance may be smaller than that of the cause (even) if both distances are measured in centimetres. Nonetheless, it may be possible to exploit varsortability if one can establish that certain assumptions on the data scale be met.

## 3.3 Varsortability in Benchmarking Scenarios

For real-world data we cannot readily assess nor presume varsortability as we do not know the parameters and data scale of the data generating process. When benchmarking causal structure learning algorithms, however, we can evaluate varsortability for the simulated DAGs and parameter settings. We may acquire an intuition about the probability of varsortable cause-effect pairs in our simulation settings by considering two neighboring nodes $A \xrightarrow{w} B$ in the sampled graph without common ancestors and no other directed path from $A$ to $B$. Under these assumptions, $\text{Var}(B) = w^2 \text{Var}(A) + \text{Var}(\sum_{C \in \text{PA}(B) \setminus \{A\}} w^2_{C \to B} C) + \text{Var}(N_B)$ such that $|w| > 1$ implies that the variable pair is varsortable. To simulate an ANM, we need to sample a DAG, decide on the noise distributions, and then sample edge weights and noise variances. Across simulation instances in a given benchmark set-up, the edge weights $w_{k \to j}$ and noise variances $s_k^2$ are iid instantiations of independent random variables $W$ and $S^2$. The distributions of $W$ and $S^2$ induce a distribution of the marginal variance $V_Y$ of node $Y$ in the resulting ANM. The probability for the variable pair $A \to B$ to be varsortable in a simulated ANM is then bounded from below by $\text{P}[(1 - W^2_{A \to B})V_A < S^2_{N_B}]$ (cf. Appendix B). If $A$ is a root node, $V_A = S^2_{N_A}$. In the experimental settings used by, for example, Zheng et al. [2018, 2020], Lachapelle et al. [2019], Ng et al. [2020], edge weights are independently drawn from a uniform distribution and noise standard deviations or variances are either fixed or also drawn independently from a uniform distribution. For our parameters $W \stackrel{\text{iid}}{\sim} \text{Unif}((-2, -0.5) \cup (0.5, 2))$ and $S \stackrel{\text{iid}}{\sim} \text{Unif}((0.5, 2))$, which resemble common choices in the literature, any pair is varsortable with probability at least $2/3$ due to $\text{P}[|W| > 1] = 2/3$, and with probability $p > 0.93$ provided $A$ is a

root node. Empirically, we find that varsortability averages above $0.94$ in our simulated graphs and above $0.71$ in commonly considered non-linear ANMs (cf. Appendix G). This result indicates that in benchmark simulations the marginal variance of any two nodes in a graph tends to increase along the causal order and that we may game these benchmarks and perform well by exploiting this pattern.

If $A$ and $B$ have a common ancestor or mediator $C$, the effect of $C$ on $B$ may either compound or partially cancel out the effect of $A$ on $B$. In practice, the effect commonly increases the variance of the effect node $B$, which may be attributed to the independent sampling of path coefficients which also renders faithfulness violations improbable [Meek, 1995]. We find varsortability to increase with graph density and the lower bound presented above to be loose. Motivated by the strong impact of different levels of varsortability on some structure learning algorithms as reported in Section 4 and Appendix H.2, we advocate an empirical evaluation and reporting of varsortability (cf. Appendix G.4 for the implementation) when simulating ANMs. We emphasize that even for varsortability $< 1$, where the order of increasing variance does not perfectly agree with the causal order, experimental results may still be largely driven by the overall agreement between increasing marginal variance and causal order. The extent to which varsortability may distort experimental comparisons of structure learning algorithms on linear ANMs is demonstrated in Section 4.

### 3.4 Marginal Variance yields Asymmetric Gradients for Causal and Anti-Causal Edges

We explain how varsortability may dominate the performance of continuous structure learning algorithms. We do not expect combinatorial structure learning algorithms that use a score-equivalent (see e.g. Yang and Chang [2002], Chickering [2002a]) criterion or scale-independent (conditional) independence tests to be dependent on the data scale. This includes *PC*, as local constraint-based algorithm, *FGES* as locally greedy score-based search using a score-equivalent criterion, and *DirectLiNGAM*, a procedure minimizing residual dependence. By contrast, combinatorial algorithms with a criterion that is not score-equivalent (such as the MSE) depend on the data scale. Due to the optimization procedure, continuous structure learning algorithms may depend on the data scale irrespective of whether the employed score is score-equivalent (as, for example, *GOLEM* for Gaussian models) or not (as, for example, *GOLEM* under likelihood misspecification or *NOTEARS*).

We first establish how varsortability affects the gradients of MSE-based score functions (which are akin to assuming equal noise variances in the Gaussian setting) and when initialising with the empty graph $0_{d \times d}$ (as is done in *NOTEARS* and *GOLEM*). Full statements of objective functions and respective gradients are found in Appendix E. Since $\nabla \mathrm{MSE}(W) \propto \mathbf{X}^\top (\mathbf{X} - \mathbf{X}W)$ we have that $\nabla \mathrm{MSE}(0_{d \times d}) \propto \mathbf{X}^\top \mathbf{X}$ and $\nabla \mathcal{L}_{EV}(0_{d \times d}) \propto {}^{1}/\|\mathbf{x}\|_2^2 \mathbf{X}^\top \mathbf{X}$. The initial gradient step of both *NOTEARS* and *GOLEM-EV* is symmetric. We have $\nabla \mathrm{MSE}(W) \propto [\mathbf{X}^\top (x_1 - \mathbf{X}w_1), ..., \mathbf{X}^\top (x_d - \mathbf{X}w_d)]$ where the $j^{\text{th}}$ column $\mathbf{X}^\top (x_j - \mathbf{X}w_j)$ reflects the vector of empirical covariances of the $j^{\text{th}}$ residual vector $x_j - \mathbf{X}w_j$ with each $x_i$. Provided a small identical step size is used across all entries of $W$ in the first step (as, for example, in *GOLEM-EV*), we empirically find the residual variance after the first gradient step to be larger in those components that have higher marginal variance (see Appendix E.3 for a heuristic argument). We observe that during the next optimization steps $\nabla \mathrm{MSE}(W)$ tends to be larger magnitude for edges pointing in the direction of nodes with high-variance residuals (which tends to be those with high marginal variance) than for those pointing in the direction of nodes with low-variance residuals (which tends to be those with low marginal variance). Intuitively, when cycles are penalized, the insertion of edges pointing to nodes with high residuals is favored as a larger reduction in MSE may be achieved than by including the opposing edge. Given high varsortability, this corresponds to favoring edges in the causal direction. This way, the global information about the causal order in case of high varsortability is effectively exploited.

Once we allow for unequal noise variances as in *GOLEM-NV*, the marginal variances lead the gradients differently. Letting $\mathrm{MSE}_j(w_j) = \frac{1}{n}\|x_j - \mathbf{X}w_j\|_2^2$, we have

$$\nabla \mathcal{L}_{NV}(W) \propto \left[ \frac{\mathbf{X}^\top (x_j - \mathbf{X}w_j)}{\mathrm{MSE}_j(w_j)} \right]_{j=1,...,d}$$

such that the logarithmic derivative breaks the symmetry of the first step for the non-equal variance formulation of *GOLEM* and we have $\nabla \mathcal{L}_{NV}(0_{d \times d}) \propto \mathbf{X}^\top \mathbf{X} \, \mathrm{diag}(\|\mathbf{x}_1\|_2^{-2} \ldots, \|\mathbf{x}_d\|_2^{-2})$. While $\nabla_W \mathrm{MSE}(W) \propto [\mathbf{X}^\top (x_j - \mathbf{X}w_j)]_{j=1,...,d}$ tends to favor edges in causal direction (see above), the column-wise inverse MSE scaling of $\mathbf{X}^\top (x_j - \mathbf{X}w_j)$ by $\mathrm{MSE}_j(w_j)$ (the residual variance in

the $j^{\text{th}}$ component) leads to larger-magnitude gradient steps for edges pointing in the direction of low-variance nodes rather than high-variance nodes. Given high varsortability, this corresponds predominantly to the anti-causal direction.

We conjecture that the first gradient steps have a dominant role in determining the causal structure, even though afterwards the optimization is governed by a non-trivial interplay of optimizer, model fit, constraints, and penalties. For this reason we focus on the first optimization steps to explain a) why continuous structure learning algorithms that assume equal noise variance work remarkably well in the presence of high varsortability and b) why performance changes once data is standardized and the marginal variances no longer hold information about the causal order. Because of the acyclicity constraint, it may be enough for a weight $w_{i \to j}$ to be greater in magnitude than its counterpart $w_{j \to i}$ early on in the optimization for the smaller edge to be pruned from there on. For a discussion of the interplay between sparsity penalties and data scale see Appendix J, which indicates that the nodes need to be on a comparable data scale for $l^1$-penalization to be well calibrated. Ng et al. [2020] provide further discussion on sparsity and acyclicity constraints in continuous DAG learning.

### 3.5  *sortnregress*: A Diagnostic Tool to Reveal Varsortability

We propose an algorithm *sortnregress* performing the following two steps:

**order search**  Sort nodes by increasing marginal variance.

**parent search**  Regress each node on all of its predecessors in that order, using a sparse regression technique to prune edges [Shojaie and Michailidis, 2010]. We employ Lasso regression [Tibshirani, 1996] using the Bayesian Information Criterion [Schwarz, 1978] for model selection.

As a baseline, *sortnregress* is easy to implement (cf. Appendix H.1) and highlights and evaluates to which extent the data scale is informative of the causal structure in different benchmark scenarios. An extension for non-linear additive noise models is obtained by using an appropriate non-linear regression technique in the parent search step, possibly paired with cross-validated recursive feature elimination. It facilitates a clear and contextualized assessment of different structure learning algorithms in different benchmark scenarios. The relationship between varsortability and the performance of sortnregress in a linear setting is shown in Appendix H.2. Varying degrees of varsortability and performance of *sortnregress* add an important dimension which current benchmarks do not consider.

## 4   Simulations

We compare the performance of the algorithms introduced in Section 2.3 on raw and standardized synthetic data. In our comparison, we distinguish between settings with different noise distributions, graph types, and graph sizes. Our experimental set-up follows those in Zheng et al. [2018], Ng et al. [2020] and we contribute results obtained repeating their experiments in Appendix K. We complement our and previous DAG-recovery results by additionally evaluating how well the DAG output by continous structure learning algorithms identifies the MEC of the ground-truth DAG.

### 4.1   Data Generation

We sample Erdös-Rényi (ER) [Erdős and Rényi, 1960] and Scale-Free (SF) [Barabási and Albert, 1999] graphs and the parameters for ANMs according to the simulation details in Table 1. For a graph specified as ER-$k$ or SF-$k$ with $d$ nodes, we simulate $dk$ edges. For every combination of parameters, we create a raw data instance and a standardized version that is de-meaned and re-scaled to unit variance. On standardized data, we have varsortability $v = \frac{1}{2}$ and the marginal variances hold no information about the causal ordering of the nodes. In all our experimental settings, varsortability averages above $0.94$ on the raw data scale (cf. Appendix G.1).

Table 1: Parameters for synthetic data generation.

| Repetitions | 10 | Edge weights | iid $\text{Unif}((-2, -.5) \cup (.5, 2))$ |
|---|---|---|---|
| Graphs | ER-2, SF-2, SF-4 | Noise distributions | Exponential, Gaussian, Gumbel |
| Nodes | $d \in \{10, 30, 50\}$ | Noise standard deviations | 1 (Gaussian-EV); iid $\text{Unif}(.5, 2)$ (others) |
| Samples | $n = 1000$ | | |

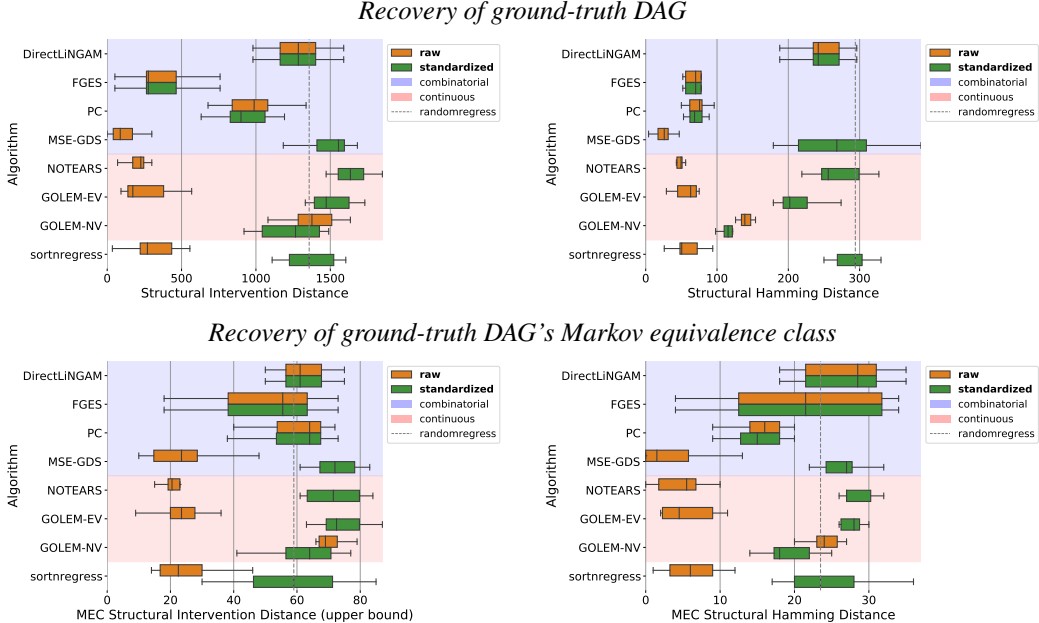

Figure 1: SID (left, lower is better) and SHD (right, lower is better) between recovered and ground-truth graphs (top) or Markov equivalence classes (bottom) for ER-2 graphs with 50 (top) or 10 (bottom) nodes and Gaussian-NV noise. The performance of *sortnregress*, which only exploits varsortability, matches that of the continuous methods *NOTEARS* and *GOLEM*.

## 4.2 Evaluation

We evaluate performance using structural intervention distance (SID) [Peters and Bühlmann, 2015] and structural Hamming distance (SHD) between the recovered graph and ground truth. Additionally, we contribute a performance assessments of continuous structure learning algorithms in terms of the SID and SHD between the ground-truth MEC and the recovered MEC (PC, FGES) or the MEC identified by the recovered graph (NOTEARS, GOLEM). SID assesses in how far the recovered structure enables to correctly predict the effect of interventions. SHD measures the distance between the true and recovered graph by counting how many edge insertions, deletions, and reversals are required to turn the former into the latter. Since interventional distributions can consistently be estimated given only the causal ordering [Bühlmann et al., 2014], SID is less susceptible to arbitrary choices of edge pruning procedures and thresholds than SHD. Intuitively, SID prioritizes the causal order, while SHD prioritizes the correctness of individual edges. We follow common practices for edge thresholding and scoring (see Appendix K).

## 4.3 Performance on Raw Versus Standardized Data

We group our algorithms into combinatorial and continuous algorithms. We propose a novel baseline algorithm termed *sortnregress* which serves as a reference marking the performance achievable by directly exploiting marginal variances. We indicate its performance on standardized data as *randomregress*, since it amounts to choosing a random order and regressing each node on its predecessors. We use boxplots aggregating the performance achieved in the 10 repetitions on raw and standardized data by each algorithm and create separate plots per noise type to account for identifiability and MLE specification differences. We show the results obtained for ER-2 graphs and Gaussian noise with non-equal variances in Figure 1. These results are representative of the results obtained for different graphs and noise distributions (cf. Appendix K). For the simulated settings, varsortability is high (> 0.94) on the raw data scale (cf. Appendix G.1).

We observe that some algorithms are highly scale-sensitive and perform vastly different on raw and standardized data. The algorithms *NOTEARS*, *MSE-GDS*, *GOLEM-EV* are most affected – their performance is excellent on raw data but far worse on standardized data. Note that all of these rely on

a loss function that revolves around the MSE. The performance of *GOLEM-NV* is also scale-sensitive but improves upon standardization. The direction of the effect of standardization is in line with the predictions by our gradient analysis in Section 3.4. Note that we initialize all algorithms with the empty graph since we are primarily interested in comparing the impact of standardization given equal starting conditions. On standardized data, an initialization of *GOLEM-NV* with the results of *GOLEM-EV*, as recommended by Ng et al. [2020], does not improve performance and may fail to converge. *sortnregress* achieves competitive performance on raw, and baseline performance on standardized data. It thus qualifies as diagnostic tool to highlight how much of a given causal structure learning task can be resolved by exploiting data scale and sorting nodes by their marginal variance.

In summary, the evidence corroborates our claim that the remarkable performance on raw data and the overall behavior upon standardization of the continuous structure learning algorithms may be driven primarily by high varsortability. On a real-world causal protein signaling dataset [Sachs et al., 2005] we measure a mean varsortability of 0.57 (which is close to chance level at 0.5) with a standard deviation of 0.01 across our bootstrapped samples and do not observe the consistent performance pattern described for synthetic data with high varsortability (cf. Appendix I).

## 5 Gaming Further Benchmarks

### 5.1 Orienting Causal Chains on Standardized Data

In order to design a causal discovery benchmark that does not favor methods that explicitly exploit marginal variances we may standardize the data or employ coefficient re-scaling schemes. Mooij et al. [2020], for example, propose a scale-harmonization by dividing each column $w_j = [w_{k \to j}]_{j=1,...,d} \in \mathbb{R}^d$ of the drawn adjacency matrices by $\sqrt{\|w_j\|^2 + 1}$ such that each variable would have comparable scale if all its direct parents were independently standard-normally distributed. However, this does not avoid the problem of potentially inadvertent patterns in simulated ANMs. Even after standardization or scale-harmonization, DAGs with previously high varsortability generate data with distinct covariance patterns that may be exploited.

In Appendix F we present an instructive example of a decision rule that can infer the orientation of a causal chain from raw, standardized, and scale-harmonized data with accuracy strictly greater than 50%. For a causal chain $X_1 \to X_2 \to ... \to X_d$ where edge weights and noise terms are drawn iid we can decide between the two Markov-equivalent graphs $X_1 \to X_2 \to ... \to X_d$ and $X_1 \leftarrow X_2 \leftarrow ... \leftarrow X_d$ whith above-chance accuracy. The empirical results for varying chain-lengths and various edge-weight distribution are deferred to the appendix where we discuss the 3-variable chain in detail and illustrate that the phenomenon extends from finite-sample to the population setting.

The intuition is as follows. Consider data generated by $X_1 \to X_2 \to ... \to X_d$ and the aim is to infer from standardized data whether $X_1 \to ... \to X_d$ or $X_1 \leftarrow ... \leftarrow X_d$. For data with high varsortability and comparable noise variance on the raw data scale it holds that the further downstream a node $X_i$ is in the causal chain, the stronger the variance of its parent $\mathrm{Var}(X_{i-1})$ contributes to its marginal variance $\mathrm{Var}(X_i) = \mathrm{Var}(X_{i-1}) + \mathrm{Var}(N_i)$ relative to its noise variance $\mathrm{Var}(N_i)$, and the stronger is it correlated with its parent. Thus, the sequence of regression coefficients, which in the standardized case amounts to $(\mathrm{Corr}(X_i, X_{i+1}))_{i=1,...,d-1}$, tends to increase in magnitude along the causal order and decrease in the anti-causal direction. The proposed decision rule predicts the causal direction as the one in which the absolute values of the regression coefficients tend to increase. This chain orientation rule achieves above-chance performance on raw, standardized, and scale-harmonized data (cf. Appendix F).

### 5.2 Sorting by Variance in Non-Linear Settings

Varsortability may also be exploited in non-linear settings. Table 2 shows the results of sorting by marginal variance and filling in all edges from lower-variance nodes to higher-variance nodes in a non-linear setting. This variance sorting strategy is more naive than *sortnregress* and places no assumption on the functional form. The results are substantially better than random sorting and may therefore be a more informative baseline than commonly used random graphs. We do not show performance in terms of SHD, as our variance sorting baseline always yields a fully connected graph. Although the data generating process is not identical, we note that the improvement of our crude variance sorting over random sorting compares favorably to some of the improvements gained by

more involved methods over random graphs as shown in Lachapelle et al. [2019, Table 1]. Our results indicate that exploiting varsortability may also deliver competitive results in non-linear settings.

Table 2: SID of naive baselines on non-linear data. Results on 1000 observations of additive Gaussian process ANMs with noise variance 1 simulated as by Zheng et al. [2020] (10 repetitions each; average varsortability *v* per graph type shown in parentheses).

| Algorithm | Graph (average *v*) | ER-1 (0.87) | ER-4 (0.95) | SF-1 (0.95) | SF-4 (0.98) |
|---|---|---|---|---|---|
| variance sorting | | $7.7 \pm 5.72$ | $25.2 \pm 12.36$ | $1.9 \pm 2.28$ | $7.6 \pm 3.37$ |
| random sorting | | $27.9 \pm 11.44$ | $63.1 \pm 8.10$ | $22.3 \pm 13.14$ | $59.5 \pm 7.32$ |

We find similarly high levels of varsortability for many non-linear functional relationships and graph parameters (cf. Appendix G.2). This begs the question how much other successful methods exploit varsortability, how they compare to non-linear nonparametric methods that leverage assumptions on the residual variances [Gao et al., 2020], and how they perform under data standardization. We encourage such an exploration in future work and suggest that varsortability and *sortnregress* or *variance sorting* should always be included in future benchmarks.

## 6 Discussion and Conclusion

We find that continuous structure learning methods are highly susceptible to data rescaling and some do not perform well without access to the true data scale. Therefore, scale-variant causal structure learning methods should be applied and benchmarked with caution, especially if the variables do not share a measurement scale or when the true scale of the data is unattainable. It is important to declare whether data is standardized prior to being fed to various structure learning algorithms.

Following the first release of the present paper, Kaiser and Sipos [2021] also independently reported the drop in performance of *NOTEARS* upon standardizing the data and presented a low-dimensional exemplary case. Beyond a reporting of impaired *NOTEARS* performance, we also analyze score-equivalent methods, provide exhaustive simulation experiments, and explain the phenomenon.

Our aim is to raise awareness of the severity with which scaling properties in data from simulated DAGs and causal additive models may distort algorithm performance. Increasing marginal variances can render scenarios identifiable, which may commonly not be expected to be so—for example the Gaussian case with non-equal variances. We therefore argue that varsortability should be taken into account for future benchmarking. Yet, with any synthetic benchmark there remains a risk that the results are not indicative of algorithm performance on real data. Our results indicate that current structure learning algorithms may perform within the range of naive baselines on real-world datasets.

The theoretical results of our paper are limited to the setting of linear ANMs. Additionally, our conjecture regarding the importance of the first gradient steps, and with it a rigorous causal explanation for the learning behavior of different continuous algorithms and corresponding score functions remain open and require further research to be settled. Our empirical findings indicate that causal discovery benchmarks can be similarly gamed on standardized data and in non-linear settings, but further research is needed to confirm this. We focus on a specific subset of algrorithms, the impact of patterns in benchmarking data on a wider class of algorithms and score functions remains to be explored.

Varsortability arises in many ANMs and the marginal variances increase drastically along the causal order, at least in common simulation settings. This begs the question what degree of varsortability can be observed or assumed in real-world data. If the marginal variances carry information about the causal order, our results suggest that it can and should be leveraged for structure learning. Otherwise, our contribution motivates future research into representative benchmarks and may put the practical applicability of the additive noise assumption into question.

### Acknowledgements

We thank Jonas M. Kübler, Jonas Peters, and Sorawit Saengkyongam for helpful discussions and comments. SW was supported by the Carlsberg Foundation.

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
