# Appendix

## Table of Contents

## A    Varsortability in the Two-Node Case

Consider the following ground truth and two competing linear acyclic models:

Ground-truth model:                Model *M1)* "$A \rightarrow B$":                Model *M2)* "$A \leftarrow B$":

$$A := N_A$$                          $$\widehat{A} = 0$$                          $$\widehat{A} = \widehat{v}B$$
$$B := wA + N_B$$                     $$\widehat{B} = \widehat{w}A$$                $$\widehat{B} = 0$$

where $w \neq 0$, $N_A$ and $N_B$ are independent zero-centred noise terms that follow some distributions with non-vanishing corresponding variance $V_A$ and $V_B$. The model parameters $\widehat{w} = \frac{\text{Cov}(A,B)}{\text{Var}(A)} = w$ and $\widehat{v} = \frac{\text{Cov}(A,B)}{\text{Var}(B)} = \frac{wV_A}{\text{Var}(B)}$ are the corresponding ordinary least-squares linear regression coefficients.

We evaluate in which cases the true model *M1* obtains a smaller MSE than the wrong model *M2*, to decide if and under which conditions a MSE-based orientation rule recovers the ground-truth edge direction:

$$\text{MSE}\,(M1) < \text{MSE}\,(M2)$$
$$\Longleftrightarrow \qquad \text{Var}(A) + \text{Var}(B - \widehat{w}A) < \text{Var}(A - \widehat{v}B) + \text{Var}(B)$$
$$\Longleftrightarrow \quad \text{Var}(A) + \text{Var}\left((wN_A + N_B) - wN_A\right) < \text{Var}\left(N_A - \frac{wV_A}{\text{Var}(B)}(wN_A + N_B)\right) + \text{Var}(B)$$
$$\Longleftrightarrow \qquad V_A + V_B < \frac{V_A V_B}{\text{Var}(B)} + w^2 V_A + V_B$$
$$\Longleftrightarrow \qquad 0 < V_A \left(\frac{V_B}{\text{Var}(B)} - 1\right) + w^2 V_A$$
$$\Longleftrightarrow \qquad 0 < \frac{-w^2 V_A}{\text{Var}(B)} + w^2$$
$$\Longleftrightarrow \qquad V_A < \text{Var}(B)$$
$$\Longleftrightarrow \qquad (1 - w^2)V_A < V_B$$

For error variances $V_A \leq V_B$ and any non-zero edge weight $w$, the MSE-based inference is correct. This resembles known scale-based identifiability results based on equal or monotonically increasing error variances [Peters and Bühlmann, 2014, Park, 2020].

## B    Derivation of Lower Bound on Pairwise Varsortability

Let $A$ and $B$ be any two nodes in the sampled graph with edge $A \overset{w}{\rightarrow} B$, noise terms $N_A, N_B$, and without common ancestors and no other directed path from $A$ to $B$. When sampling edge coefficients and noise variances randomly for the simulation of ANMs, distributions are incurred over the variances of $A$ and $B$ across those simulated ANMs. Let edge weights be sampled as $[W_{x \rightarrow y}]_{x,y=1,\ldots,d} \sim \mathbb{P}_W$, and noise variances be sampled as $[S^2_{N_y}]_{y=1,\ldots,d} \sim \mathbb{P}_{S^2}$. Across simulations, the marginal variances of $A$ and $B$ are transformations of $S$ and $W$ and themselves random variables denoted as $V_A$ and $V_B$. The marginal variance $V_Y$ of any node $Y$ depends on its noise variance and the additional variance incurred by predecessor nodes given as $\sum_{X \in \text{PA}(Y)} W^2_{X \rightarrow Y} V_X$. We can therefore bound the probability for the variable pair $(A, B)$ to be varsortable from below via

$$\text{P}[V_A < V_B] = \text{P}\left[V_A < \left(W^2_{AB}V_A + \sum_{X \in \text{PA}(B)\backslash\{A\}} W^2_{XB}V_X + S^2_{N_B}\right)\right]$$
$$\geq \text{P}[V_A < W^2_{AB}V_A + S^2_{N_B}]$$

where equality holds if $A$ is the only parent of $B$ contributing to $B$'s marginal variance.

In common benchmarks, edge weights are drawn independently according to $\mathbb{P}_W \sim \otimes_{k,j=1,\dots,d} \mathrm{Unif}((-2,-.5) \cup (.5,2))$ and noise standard deviations are drawn iid $S_{N_j} \sim \mathrm{Unif}(.5,2)$.

## C   Varsortability and Identifiability by Conditional Noise Variances

While closely related, varsortability is not equivalent to the identifiability conditions laid out in Theorem 4, Park [2020], (henceforth referred to as "Theorem 4"). We prove this by providing two examples. In Appendix C.1 part A) of the conditions in Theorem 4 is satisfied, while varsortability does not hold. In Appendix C.2 varsortability holds but neither part A) nor part B) of Theorem 4 are satisfied.

### C.1   Park, 2020, Theorem 4 conditions satisfied without varsortability

Consider the following ground-truth model with unique causal order $A, B, C$:

$$A := N_A$$
$$B := \beta_{A \to B} A + N_B = 1A + N_B$$
$$C := \beta_{B \to C} B + N_C = \sqrt{\frac{2}{3}} B + N_C$$

where $N_A, N_B, N_C$ are jointly independent zero-centred noise terms with respective variances $\sigma_A^2 = 4, \sigma_B^2 = 2, \sigma_C^2 = 1$. The marginal variances are $\mathrm{Var}(A) = 4 < \mathrm{Var}(C) = 5 < \mathrm{Var}(B) = 6$. Our example resembles the examples in Section 3.1 of Park [2020]. We can verify the three conditions for part A) of Theorem 4:

$$(A1) \quad \sigma_A^2 < \sigma_B^2 + \beta_{A \to B}^2 \sigma_A^2,$$
$$(A2) \quad \sigma_B^2 < \sigma_C^2 + \beta_{B \to C}^2 \sigma_B^2,$$
$$(A3) \quad \sigma_A^2 < \sigma_C^2 + \beta_{B \to C}^2 \sigma_B^2 + \beta_{A \to B}^2 \beta_{B \to C}^2 \sigma_A^2$$

Inserting the values from above, we obtain

$$(A1) \quad 4 < 2 + 1 \cdot 4, \qquad (A2) \quad 2 < 1 + \frac{2}{3} \cdot 2, \qquad (A3) \quad 4 < 1 + \frac{2}{3} \cdot 2 + 1 \cdot \frac{2}{3} \cdot 4.$$

Our result verifies that identifiability is given as per Theorem 4 in Park [2020], while the order of increasing marginal variances is not in complete agreement with the causal order and varsortability is not equal to 1.

### C.2   Varsortability without Park, 2020, Theorem 4 conditions satisfied

Consider the following ground-truth model with unique causal order $A, B, C$:

$$A := N_A$$
$$B := \beta_{A \to B} A + N_B = A + N_B$$
$$C := \beta_{A \to C} A + \beta_{B \to C} B + N_C = \frac{1}{\sqrt{2}} A + \frac{1}{\sqrt{2}} B + N_C$$

where $N_A, N_B, N_C$ are jointly independent zero-centred noise terms with respective variances $\sigma_A^2 = 4, \sigma_B^2 = 3, \sigma_C^2 = 1$. The marginal variances are $\mathrm{Var}(A) = 4 < \mathrm{Var}(B) = 7 < \mathrm{Var}(C) = 10.5$. We now verify, that for both case A) and B) in Theorem 4 of Park [2020] at least one of the inequality constraints is violated.

One of the three conditions in A) is

$$\sigma_B^2 < \sigma_C^2 + \beta_{B \to C}^2 \sigma_B^2,$$

while for the above model we have

$$3 \not\prec 1 + \frac{1}{2} \cdot 3.$$

One of the three conditions in B) is

$$\frac{\sigma_C^2}{\sigma_B^2} > (1 - \beta_{B \to C}^2),$$

while for the above model we have

$$\frac{1}{3} \not> (1 - \frac{1}{2}).$$

For both criteria A) and B) in Theorem 4 at least one of the inequalities is not satisfied. We thus verify that even if identifiability is not given as per the sufficient conditions in Theorem 4, Park [2020], varsortability may still render the causal order identifiable.

## D  Algorithms

**DirectLiNGAM**  is a method for learning linear non-Gaussian acyclic models [Shimizu et al., 2011]. It recovers the causal order by iteratively selecting the node whose residuals are least dependent on any predecessor node. In a strictly non-Gaussian setting, *DirectLiNGAM* is guaranteed to converge to the optimal solution asymptotically within a small fixed number of steps and returns a DAG. We use the implementation provided by the authors[1]. We deliberately keep the default of a least-angle regression penalized by the Bayesian Informaion Criterion. We find that this penalty strikes a good balance between SID and SHD performance. Cross-validated least-angle regression performs better in terms of SID but poorer in terms of SHD.

**PC**  [Spirtes and Glymour, 1991] is provably consistent in estimating the Markov equivalence class of the true data-generating graph if the causal Markov and faithfulness assumptions hold. The algorithm returns a completed partially directed acyclic graph (CPDAG). For computational reasons, we refrain from computing the lower and upper bounds of the SID for comparing CPDAGS with the ground-truth DAG as proposed by Peters and Bühlmann [2015]. Instead, we adopt the approach by Zheng et al. [2018] and resolve bidirectional edges favorably to obtain a DAG. We use the implementation in the *Tetrad*[2] package Ramsey et al. [2018].

**FGES**  is an optimized version of the fast greedy equivalence search algorithm developed by Meek [1997], Chickering [2002b]. Under causal Markov and faithfulness assumptions, it is provably consistent for estimating the Markov equivalence class of the true data-generating graph. The algorithm returns a CPDAG, which we resolve favorably to obtain a DAG. We use the implementation in the *Tetrad*[3] package [Ramsey et al., 2018].

**MSE-GDS**  is a greedy DAG search procedure with a MSE score criterion. We implement *MSE-GDS* following other GDS procedures, for example, as described by Peters and Bühlmann [2014, Section 4], but use the MSE as score criterion instead of a likelihood- or BIC-based score criterion. For simplicity and computational ease, we consider a smaller search space and greedily forward-search over new edge insertions only instead of greedily searching over all neighbouring DAGs obtainable by edge insertions, removals, and deletions. For the linear setting, linear regression is used to determine the edge weights and the corresponding MSE-score for a given graph. For the non-linear setting, support vector regression can be used instead. The algorithm returns a DAG.

**NOTEARS**  is a score-based method that finds both structure and parameters simultaneously by continuous optimization [Zheng et al., 2018]. The optimization formulation is based on the mean squared error and includes a sparsity penalty parameter $\lambda$ and a differentiable acyclicity constraint:

$$\underset{W \in \mathbb{R}^{d \times d}}{\operatorname{argmin}} \quad \text{MSE}_{\mathbf{X}}(W) + \lambda \|W\|_1 \quad \text{s.t.} \quad \text{tr}(\exp(W \odot W)) - d = 0.$$

---

[1]https://github.com/cdt15/lingam
[2]https://github.com/cmu-phil/tetrad
[3]https://github.com/cmu-phil/tetrad

The algorithm returns a DAG. We use the implementation provided by the authors[4]. Throughout all our experiments we use *NOTEARS* over *NOTEARS-L1* (setting $\lambda = 0$), following the findings of Zheng et al. [2018, Tables 1 and 2, Figure 3], which suggest regularization only for samples smaller than the $n = 1000$ we use throughout.

**GOLEM** combines a soft version of the differentiable acyclicity constraint from Zheng et al. [2018] with a MLE objective [Ng et al., 2020]. The authors propose a multivariate Gaussian MLE for equal (EV) or unequal (NV) noise variances and optimize

$$\operatorname*{argmin}_{W \in \mathbb{R}^{d \times d}} \; \widetilde{\mathcal{L}}(W, \mathbf{X}) - \log(|\det(I - W)|) + \lambda_1 \|W\|_1 + \lambda_2(\operatorname{tr}(\exp(W \odot W)) - d)$$

where $\widetilde{\mathcal{L}}$ is either

$$\widetilde{\mathcal{L}}_{EV}(W, \mathbf{X}) = \frac{d}{2}\left(\mathcal{L}_{EV}(W, \mathbf{X}) + \log(n)\right) = \frac{d}{2}\log(n\operatorname{MSE}_{\mathbf{X}}(W)), \text{ or}$$

$$\widetilde{\mathcal{L}}_{NV}(W, \mathbf{X}) = \frac{1}{2}\left(\mathcal{L}_{NV}(W, \mathbf{X}) + d\log(n)\right) = \frac{1}{2}\sum_{j=1}^{d}\log(n\operatorname{MSE}_j(w_j)).$$

We use the implementation and hyperparameters provided by the authors[5]. We train for $10^4$ episodes as we found that half of that suffices to ensure convergence. Notably, we do not perform pretraining for our version of *GOLEM-NV*.

**sortnregress** is implemented as shown in Appendix H.1. We find that a least-angle regression penalized by the Bayesian Information Criterion strikes a good balance between SID and SHD performance.

# E   The Subtle Interplay Between Marginal Variance and Gradient Directions

We describe observations about the gradients involved in the optimization procedures of *NOTEARS* and *GOLEM-EV/-NV*. We present an instructive example in Appendix E.1 and provide some intuition about how the adjacency matrix changes throughout the optimization. For convenience and reference we provide gradients of the individual terms involved in the respective objective functions (cf. Appendix E.2). In Appendix E.3 we argue why the nodes' residual variances for the first gradient steps in an unconstrained optimization of MSE- or log-MSE-EV-based (GOLEM-EV) objective functions with acyclicity penalties tend to follow the same ordering as the nodes' marginal variances. We analyze gradient symmetry and asymmetry in *GOLEM-EV/-NV*'s gradient descent optimization under varsortability in Appendix E.4. While the intuition for small step size gradient-based unconstrained optimization partially carries over to the *NOTEARS* optimization procedure, here the interplay between varsortability and gradient directions is intricate due to a constrained optimization that is solved via the augmented Lagrangian method and dual descent with line-search instead of gradient descent as used in *GOLEM* [Zheng et al., 2018] (cf. Appendix E.5).

The heuristic arguments presented here are preliminary and aim to provide intuition. The optimization behaviour also heavily depends on the implementation of the optimization routine. For example, the original implementation of *NOTEARS* fixes the diagonal of $W$ at zero and leverages curvature information (L-BFGS-B), while *GOLEM* updates all entries of $W$ and employs learning rate optimizers. Future research is required to determine how precisely continuous structure learning algorithms achieve state-of-the-art results on highly varsortable data and, given our observations, we expect explanations to be specific to individual algorithms and their distinct implementations.

## E.1   Example

The following example considers the population limit and illustrates a few intuitions about gradient based optimization and varsortability. Consider data is generated according to

$$\begin{pmatrix} X \\ Y \end{pmatrix} = \begin{pmatrix} 0 & \beta \\ 0 & 0 \end{pmatrix}^{\top} \begin{pmatrix} X \\ Y \end{pmatrix} + \begin{pmatrix} N_X \\ N_Y \end{pmatrix}$$

---

[4] https://github.com/xunzheng/notears
[5] https://github.com/ignavier/golem

where $N_X$ and $N_Y$ are independently normally distributed with standard deviations $\sigma_{N_X}$ and $\sigma_{N_Y}$. Here, varsortability $v = 1$ and $1 = \text{Var}\, X < \text{Var}\, Y = 2$.

Initializing the weight matrix at the zero matrix, the gradient of the population MSE is

$$-2 \begin{pmatrix} \text{Var}(X) & \text{Cov}(X, Y) \\ \text{Cov}(Y, X) & \text{Var}(Y) \end{pmatrix} = -2 \begin{pmatrix} \sigma_{N_X}^2 & \beta\sigma_{N_X}^2 \\ \beta\sigma_{N_X}^2 & \beta^2\sigma_{N_X}^2 + \sigma_{N_Y}^2 \end{pmatrix}$$

(see also Appendix E.2). The models for $X$ and $Y$ after a first gradient descent step of step size $\eta$ are

$$\widehat{X} = 2\eta(\sigma_{N_X}^2 X + \beta\sigma_{N_X}^2 Y)$$
$$\widehat{Y} = 2\eta(\beta\sigma_{N_X}^2 X + (\beta^2\sigma_{N_X}^2 + \sigma_{N_Y}^2)Y)$$

If the diagonal of the weight matrix is clamped to 0 throughout the optimization, the terms corresponding to self-loops ($2\eta\sigma_{N_X}^2 X$ in $\widehat{X}$ and $(\beta^2\sigma_{N_X}^2 + \sigma_{N_Y}^2)Y$ in $\widehat{Y}$) are dropped above. This is the case in the original implementation of NOTEARS, where the unconstrained subproblem is optimized via L-BFGS-B with identity bounds on the diagonal entries of $W$.

Below we visualize $\text{Var}(X - \widehat{X})$ (residual variance in $X$), $\text{Var}(Y - \widehat{Y})$ (residual variance in $Y$), and the MSE $\text{Var}(X - \widehat{X}) + \text{Var}(Y - \widehat{Y})$, for varying step sizes $\eta$ of the first gradient step where we exemplary choose $\beta = \sigma_{N_X} = \sigma_{N_Y} = 1$.

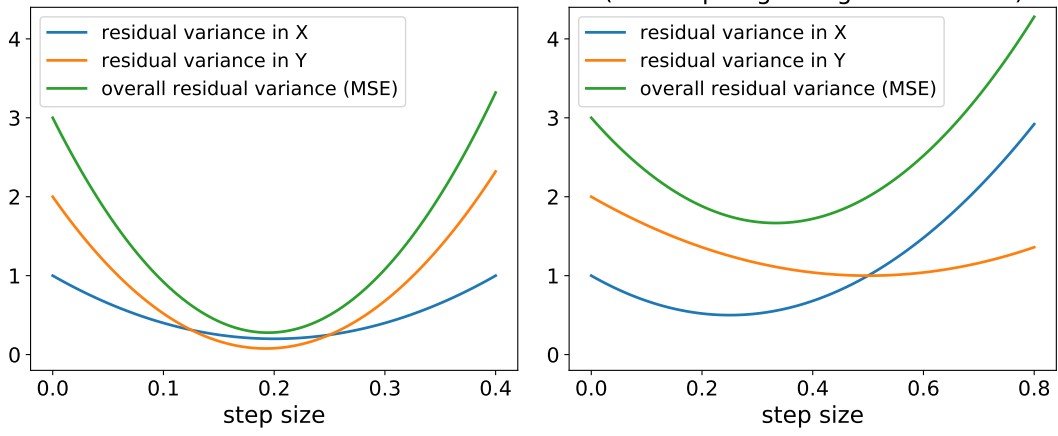

Since the residual variances change continuously for increasing step sizes, the residual variances follow the order of the marginal variances for small step sizes (cf. also Appendix E.3). Since in GOLEM we solve an unconstrained optimization problem by gradient descent (with small step size and learning rate), the order of residual variances tends to remain unchanged during the first optimization steps. The order of the residual variances may swap relative to the order of marginal variances, though, if line-search is employed to determine the step size that minimizes the MSE-objective. This is the case in NOTEARS, where the MSE is minimized by a dual descent routine with increasing weight on the acyclicity penalty term. Here, the first symmetric update of the weight matrix occurs with a large step size that minimizes the MSE (minimum of the green curves in above plots). The ordering of the resulting residual variances is less obvious. In the above example, if the diagonal terms of the weight matrix are updated as well (left), the residual variance order after the first gradient step is opposite to the marginal variance order. If the diagonal entries are clamped at 0 (as is the case in *NOTEARS* and corresponding to the setting shown on the right), the first gradient step in the above example leads to a scenario where the residual variance order follows the marginal variance order and where the resulting edge weight for the direction $X \leftarrow Y$ overshoots the optimum, that is, the blue curve's minimum is attained for a smaller step size than the green curve's minimum. The intuition is as follows: If we minimize the MSE the step size calibrates a trade-off between residual variances in the different nodes; the high marginal variance nodes dominate the MSE such that the step size that minimizes the MSE may result in ill-chosen edge weights for the edges incoming into low-variance nodes. In the next optimization step, the gradient of the MSE loss for the edge $X \to Y$ pushes towards increasing that edge weight, while it pushes for decreasing the edge weight

$X \leftarrow Y$ (besides a gradient contribution from the acyclicity constraint). As a result, the edge weights for $X \rightarrow Y$ and $X \leftarrow Y$ are equal after the first step of *NOTEARS*, but better calibrated for the direction from low- to high-variance nodes, which here corresponds to the correct edge $X \rightarrow Y$. In the subsequent optimization step, decreasing the edge weight $X \leftarrow Y$ is favored both by the MSE gradient and the acyclicity penalty, while for the correct edge $X \rightarrow Y$ the MSE gradient pushes to further increasing the edge. Intuitively, if one needs to cut one of the two edges to avoid cycles, it is "cheaper" in MSE to cut the wrong edge $X \leftarrow Y$ from a high- to low-variance node.

### E.2  Stepwise Gradient Derivation

**MSE**  For $\mathbf{X} \in \mathbb{R}^{n \times d}$, the gradient of $\mathrm{MSE}_{\mathbf{X}}(W) = \frac{1}{n}\|\mathbf{X} - \mathbf{X}W\|_2^2$ is

$$
\begin{aligned}
\nabla_W \mathrm{MSE}_{\mathbf{X}}(W) &= \frac{1}{n}\nabla_W \left(\mathrm{Tr}[\mathbf{X}^\top \mathbf{X}] - \mathrm{Tr}[W^\top \mathbf{X}^\top \mathbf{X}] - \mathrm{Tr}[\mathbf{X}^\top \mathbf{X}W] + \mathrm{Tr}[W^\top \mathbf{X}^\top \mathbf{X}W]\right) \\
&= \frac{1}{n}\left(-\mathbf{X}^\top \mathbf{X} - \mathbf{X}^\top \mathbf{X} + \mathbf{X}^\top \mathbf{X}W + \mathbf{X}^\top \mathbf{X}W\right) \\
&= -\frac{2}{n}\left(\mathbf{X}^\top \mathbf{X} - \mathbf{X}^\top \mathbf{X}W\right) \\
&= -\frac{2}{n}\mathbf{X}^\top(\mathbf{X} - \mathbf{X}W) \\
&\propto \mathbf{X}^\top(\mathbf{X} - \mathbf{X}W)
\end{aligned}
$$

If $W$ is polynomial in $\mathbf{X}^\top \mathbf{X}$, $\nabla_W \mathrm{MSE}_{\mathbf{X}}(W)$ is symmetric. $\nabla_W \mathrm{MSE}_{\mathbf{X}}(\mathbf{0}_{d \times d}) = -\frac{2}{n}\mathbf{X}^\top \mathbf{X}$.

**GOLEM-EV**  The gradient of the unnormalized negative likelihood-part of the *GOLEM-EV* objective denoted as $\widetilde{\mathcal{L}}_{EV}(W, \mathbf{X})$ is

$$
\begin{aligned}
\nabla_W \widetilde{\mathcal{L}}_{EV}(W, \mathbf{X}) &= \frac{d}{2}\nabla_W \log(n\,\mathrm{MSE}_{\mathbf{X}}(W)) \\
&= \frac{d}{2}\frac{1}{\mathrm{MSE}_{\mathbf{X}}(W)}\nabla_W \mathrm{MSE}_{\mathbf{X}}(W) \\
&\propto \frac{1}{\mathrm{MSE}_{\mathbf{X}}(W)}\mathbf{X}^\top(\mathbf{X} - \mathbf{X}W)
\end{aligned}
$$

If $W$ is polynomial in $\mathbf{X}^\top \mathbf{X}$, $\nabla_W \widetilde{\mathcal{L}}_{EV}(W, \mathbf{X})$ is symmetric. $\nabla_W \widetilde{\mathcal{L}}_{EV}(\mathbf{0}_{d \times d}, \mathbf{X}) = -\frac{d}{\|\mathbf{X}\|_2^2}\mathbf{X}^\top \mathbf{X}$.

**GOLEM-NV**  The gradient of the unnormalized negative likelihood-part of the *GOLEM-NV* objective denoted as $\widetilde{\mathcal{L}}_{NV}(W, \mathbf{X})$ is

$$
\begin{aligned}
\nabla_W \widetilde{\mathcal{L}}_{NV}(W, \mathbf{X}) &= \frac{1}{2}\sum_{j=1}^{d}\nabla_W \log(n\,\mathrm{MSE}_j(w_j)) \\
&= \left[-\frac{1}{n\,\mathrm{MSE}_j(w_j)}\mathbf{X}^\top(x_j - \mathbf{X}w_j)\right]_{j=1,\dots,d} \\
&\propto \left[\frac{\mathbf{X}^\top(x_j - \mathbf{X}w_j)}{\mathrm{MSE}_j(w_j)}\right]_{j=1,\dots,d}
\end{aligned}
$$

For the zero matrix, we have $\nabla_W \widetilde{\mathcal{L}}_{NV}(\mathbf{0}_{d \times d}, \mathbf{X}) = -\mathbf{X}^\top \mathbf{X} \operatorname{diag}\left(\|x_1\|_2^{-2}, \dots, \|x_d\|_2^{-2}\right)$.

We focus on the gradients of MSE, $\mathcal{L}_{EV}$, and $\mathcal{L}_{NV}$ since l1 penalty, acyclicity penalty $h$, LogDet term, and exact scaling of $\widetilde{\mathcal{L}}_{EV}$ and $\widetilde{\mathcal{L}}_{NV}$ play a subordinate role at the zero initialization, where the LogDet gradient has zero off-diagonals and $\nabla_W h$ vanishes:

**The LogDet in *GOLEM-EV* and *GOLEM-NV***  $\mathrm{LogDet}(W) = \log(\det(I - W))$ has gradient

$$
\nabla_W \mathrm{LogDet}(W) = -(I - W)^{-\top}
$$

and vanishes when $W$ is the adjacency matrix of a DAG [Ng et al., 2020]. If $W$ is symmetric, $\nabla_W \mathrm{LogDet}(W)$ is symmetric. For the zero matrix, we have $\nabla_W \mathrm{LogDet}(\mathbf{0}_{d \times d}) = -I$.

**Acyclicity Penalty/Constraint** The function $h(W) = \mathrm{tr}(\exp(W \odot W)) - d$ has gradient $\nabla_W h(W) = \exp(W \odot W)^\top \odot 2W$. The $h(W){=}0$-level set characterizes adjacency matrices of DAGs [Zheng et al., 2018]. If $W$ is symmetric, $\nabla_W h(W)$ is symmetric. For the zero matrix, we have $h(\mathbf{0}_{d\times d}) = 0$ and $\nabla_W h(\mathbf{0}_{d\times d}) = \mathbf{0}_{d\times d}$.

### E.3   Increasing Marginal and Residual Variances

We observe a strong positive correlation between the ordering by marginal variance and the ordering by residual variance after the first gradient step when minimizing a MSE- or likelihood-based objective function via gradient descent with small step size (as in *GOLEM-EV/-NV*). For small step sizes and learning rates, marginal variance order and residual variance order are perfectly aligned for the first few optimization steps. Here we argue for a MSE-based loss function why the residual variance follows the order of increasing marginal variance after the first optimisation step with sufficiently small step size. Future work may investigate subsequent optimisation steps and the non-MSE terms of the objective functions.

Consider the data matrix $\mathbf{X} \in \mathbb{R}^{n\times d}$. Without loss of generality, we assume the columns are zero-centred and ordered such that the sequence of diagonal entries $\mathrm{diag}(\mathbf{X}^\top\mathbf{X})$ is weakly monotonically increasing. The diagonal entries $\mathrm{diag}(\mathbf{X}^\top\mathbf{X})$ correspond to ($n$-times) the marginal variances at step 0. After the first gradient step with step size $\alpha$ in direction $-\nabla_W \mathrm{MSE}_{\mathbf{X}}(\mathbf{0}_{d\times d}) = \frac{2}{n}\mathbf{X}^\top\mathbf{X}$ (cf. Appendix E.2) the vector of ($n$-times) the residual variances is

$$\mathbf{R} = \mathrm{diag}\left([\mathbf{X} - a\mathbf{X}\mathbf{X}^\top\mathbf{X}]^\top[\mathbf{X} - a\mathbf{X}\mathbf{X}^\top\mathbf{X}]\right)$$
$$= \mathrm{diag}\left(\mathbf{D}\right) - 2a\,\mathrm{diag}\left(\mathbf{D}^2\right) + a^2\,\mathrm{diag}\left(\mathbf{D}^3\right)$$

where $\mathbf{D} = \mathbf{X}^\top\mathbf{X}$ and $a = \frac{2}{n}\alpha$. For each coordinate $i$ the residual variance $\mathbf{R}_i$ is a continuous function in $a$ (and $\alpha$). For $a = 0$ and every $i \in [1, ..., d-1]$ we have $\mathbf{R}_{i+1} - \mathbf{R}_i = \mathbf{D}_{i+1} - \mathbf{D}_i \geq 0$ with strict inequality if the variable pair $i, i+1$ is varsortable. Due to continuity, for any pair of variables with unequal marginal variances, there exists a sufficiently small step size to ensure that the resulting residual variances follow the same order as the marginal variances.

### E.4   Gradient Asymmetry

We combine what we laid out in Appendix E.2.

**The GOLEM-EV optimization problem is**

$$\underset{W}{\mathrm{argmin}}\ \widetilde{\mathcal{L}}_{EV}(W, \mathbf{X}) - \mathrm{LogDet}(W) + \lambda_1\|W\|_1 + \lambda_2 h(W)$$

with the following gradient of the objective function

$$-\frac{d}{n\,\mathrm{MSE}_{\mathbf{X}}(W)}\mathbf{X}^\top(\mathbf{X} - \mathbf{X}W) + (I - W)^{-\top} + \lambda_1 W \oslash |W| + \lambda_2 \exp(W \odot W)^\top \odot 2W$$

which at zero reduces to

$$-\frac{d}{\|\mathbf{X}\|_2^2}\mathbf{X}^\top\mathbf{X} + I.$$

**The GOLEM-NV optimization problem is**

$$\underset{W}{\mathrm{argmin}}\ \widetilde{\mathcal{L}}_{NV}(W, \mathbf{X}) - \mathrm{LogDet}(W) + \lambda_1\|W\|_1 + \lambda_2 h(W)$$

with the following gradient of the objective function

$$\left[-\frac{1}{n\,\mathrm{MSE}_j(w_j)}\mathbf{X}^\top(x_j - \mathbf{X}w_j)\right]_{j=1,...,d} + (I - W)^{-\top} + \lambda_1 W \oslash |W| + \lambda_2 \exp(W \odot W)^\top \odot 2W$$

which at zero reduces to

$$-\mathbf{X}^\top\mathbf{X}\,\mathrm{diag}(\|x_1\|_2^{-2}, ..., \|x_d\|_2^{-2}) + I.$$

The gradient in *GOLEM-EV* is symmetric at $\mathbf{0}_{d\times d}$ at the first gradient descent step, but not in general for later steps. The gradient in *GOLEM-NV* is in general not symmetric and at $\mathbf{0}_{d\times d}$ (at the first

gradient descent step) the gradients for edges incoming into a node are inversely scaled by its marginal variance; consequently, for weights $w_{i \to j}$ and $w_{j \to i}$ of opposing edges the first gradient step is larger magnitude for the direction with lower-variance end-note and $w_{i \to j}$ is preferred over $w_{j \to i}$ if the variance of $X_i$ is higher than that of $X_j$. Under high-varsortability, the first *GOLEM-NV* gradient step thus tends to favor edges in anti-causal direction over those in causal direction.

## E.5 NOTEARS

The *NOTEARS* optimization problem is $\operatorname{argmin}_W \frac{1}{2} \operatorname{MSE}_{\mathbf{X}}(W)$ s.t. $h(W) = 0$ which is solved via the augmented Lagrangian method and dual descent [Zheng et al., 2018] (we omit the penalty term for the NOTEARS-l1 variant). In the original implementation, the algorithm is initialized at $\mathbf{0}_{d \times d}$ and the diagonal of $W$ is not updated but fixed to zero (this amounts to dual projected descent, where the adjacency matrix is projected onto the matrices with zero diagonal at each step avoiding self-loops per fiat).

The augmented Lagrangian

$$\frac{1}{2} \operatorname{MSE}_{\mathbf{X}}(W) + \frac{\rho}{2} h(W)^2 + \alpha h(W)$$

has gradient

$$-\frac{1}{n} \mathbf{X}^\top (\mathbf{X} - \mathbf{X}W) + (\rho h + \alpha) \left( \exp(W \odot W)^\top \odot 2W \right)$$

which at zero reduces to

$$-\frac{1}{n} \mathbf{X}^\top \mathbf{X}$$

The step size of the first gradient step in direction $\propto \mathbf{X}^\top \mathbf{X}$ is optimized by line-search to minmize the overall MSE. As seen in the example in Appendix E.1, the residual variances may or may not follow the order of the marginal variances after this first step due to the step size being larger than the small step size that would ensure agreement between the orders (cf. Appendix E.3). Nonetheless, the step size optimized by line-search aims to optimize the overall MSE which tends to favor a better fit for edges incoming into nodes with high-marginal variance. As a result, the first gradient step results in edge weights that are better calibrated for edges incoming into high-marginal variance nodes than into low-marginal variance nodes. In subsequent steps of the dual ascent procedure with increasing acyclicity penalty, the reduction of overall MSE stands at odds with satisfying the DAG constraints; it is then more costly in terms of MSE to change the weights for edges into high-marginal nodes than into low-marginal nodes such that predominantly the edges into low-variance nodes tend to be removed to eventually satisfy the acyclicity constraint. Under high varsortability, this amounts to a preference for causal edges.

# F  Standardization Is Not Enough and Regression Coefficients Tend to Increase Along the Causal Order

Code to reproduce the calculations and results in this section is available at https://github.com/Scriddie/Varsortability.

## F.1  Infinite Sample

Here, we first discuss the three-variable case to complement the intuition provided in the main text. Consider the following ground-truth linear additive acyclic models, where the second model corresponds to a standardization of the first, and the third model corresponds to a re-scaled version of the first following Mooij et al. [2020]:

| Raw ground-truth model | Standardized model | Scale-harmonized model |
|---|---|---|
| $A := N_A$ | $A_s := A / \sqrt{\operatorname{Var}(A)}$ | $A_m := N_A$ |
| $B := \beta_{A \to B} A + N_B$ | $B_s := B / \sqrt{\operatorname{Var}(B)}$ | $B_m := \dfrac{\beta_{A \to B}}{\sqrt{\beta_{A \to B}^2 + 1}} A_m + N_B$ |
| $C := \beta_{B \to C} B + N_C$ | $C_s := C / \sqrt{\operatorname{Var}(C)}$ | $C_m := \dfrac{\beta_{B \to C}}{\sqrt{\beta_{B \to C}^2 + 1}} B_m + N_C$ |

where, following common benchmark sampling schemes, $N_A$, $N_B$, and $N_C$ are independent zero-centred noise terms that follow some distributions with non-vanishing standard deviations $\sigma_A$, $\sigma_B$, and $\sigma_C$ sampled independently from $\mathrm{Unif}(.5, 2)$ and where $\beta_{A \to B}$ and $\beta_{B \to C}$ are independently drawn from $\mathrm{Unif}((-2, -.5) \cup (.5, 2))$. For any two nodes $X$ and $Y$, $\beta_{X \to Y}$ denotes an underlying model parameter, while $\widehat{\beta}_{X \to Y}$ denotes the ordinary least-squares linear regression coefficient when regressing $Y$ onto $X$ which is given as $\widehat{\beta}_{X \to Y} = \frac{\mathrm{Cov}(X,Y)}{\mathrm{Var}(X)}$.

Given observations from a variable triplet $(X, Y, Z)$, the *causal chain orientation task* is to infer whether the data generating causal chain is $X \to Y \to Z$, that is, $(X, Y, Z) = (A, B, C)$ or $Z \leftarrow Y \leftarrow X$, that is, $(Z, X, Y) = (A, B, C)$. While both graphs are Markov equivalent, we can identify the correct orientation of the causal chain, for all three considered scaling regimes, with accuracy strictly greater than 50% by applying the following procedure:

**Chain orientation rule:**

- If $|\widehat{\beta}_{X \to Y}| < |\widehat{\beta}_{Y \to Z}|$ and $|\widehat{\beta}_{Z \to Y}| > |\widehat{\beta}_{Y \to X}|$, conclude $(X, Y, Z) = (A, B, C)$.

  We conclude that $X \to Y \to Z$, if the regression coefficients are increasing in magnitude when regressing pairwise from "left to right".

- If $|\widehat{\beta}_{X \to Y}| > |\widehat{\beta}_{Y \to Z}|$ and $|\widehat{\beta}_{Z \to Y}| < |\widehat{\beta}_{Y \to X}|$, conclude $(X, Y, Z) = (C, B, A)$.

  We conclude that $X \leftarrow Y \leftarrow Z$, if the regression coefficients are increasing in magnitude when regressing pairwise from "right to left".

- Otherwise, flip a coin to decide the orientation of the underlying causal chain.

For each data scale regime, we can obtain the population regression coefficients and express those in terms of the sampled model coefficients $\beta_{A \to B}, \beta_{B \to C}, \sigma_A, \sigma_B, \sigma_C$:

- Raw ground-truth model

  - "left to right": $\widehat{\beta}_{A \to B} = \beta_{A \to B}$ and $\widehat{\beta}_{B \to C} = \beta_{B \to C}$

  - "right to left": $\widehat{\beta}_{C \to B} = \dfrac{\beta_{B \to C}\left(\beta_{A \to B}^2 \sigma_A^2 + \sigma_B^2\right)}{\beta_{A \to B}^2 \beta_{B \to C}^2 \sigma_A^2 + \beta_{B \to C}^2 \sigma_B^2 + \sigma_C^2}$ and $\widehat{\beta}_{B \to A} = \dfrac{\beta_{A \to B} \sigma_A^2}{\beta_{A \to B}^2 \sigma_A^2 + \sigma_B^2}$

- Standardized model

  - "left to right":
    $$\widehat{\beta}_{A_s \to B_s} = \frac{\beta_{A \to B} \sigma_A^2}{\sqrt{\beta_{A \to B}^2 \sigma_A^2 + \sigma_B^2}\sqrt{\sigma_A^2}} \quad \text{and} \quad \widehat{\beta}_{B_s \to C_s} = \frac{\beta_{B \to C}\sqrt{\beta_{A \to B}^2 \sigma_A^2 + \sigma_B^2}}{\sqrt{\beta_{A \to B}^2 \beta_{B \to C}^2 \sigma_A^2 + \beta_{B \to C}^2 \sigma_B^2 + \sigma_C^2}}$$

  - "right to left":
    $$\widehat{\beta}_{C_s \to B_s} = \frac{\beta_{B \to C}\sqrt{\beta_{A \to B}^2 \sigma_A^2 + \sigma_B^2}}{\sqrt{\beta_{A \to B}^2 \beta_{B \to C}^2 \sigma_A^2 + \beta_{B \to C}^2 \sigma_B^2 + \sigma_C^2}} \quad \text{and} \quad \widehat{\beta}_{B_s \to A_s} = \frac{\beta_{A \to B} \sigma_A^2}{\sqrt{\beta_{A \to B}^2 \sigma_A^2 + \sigma_B^2}\sqrt{\sigma_A^2}}$$

- Scale-harmonized model

  - Regression coefficients "from left to right":
    $$\widehat{\beta}_{A_m \to B_m} = \frac{\beta_{A \to B}}{\sqrt{\beta_{A \to B}^2 + 1}} \quad \text{and} \quad \widehat{\beta}_{B_m \to C_m} = \frac{\beta_{B \to C}}{\sqrt{\beta_{B \to C}^2 + 1}}$$

  - Regression coefficients "from right to left":
    $$\widehat{\beta}_{C_m \to B_m} = \frac{\beta_{B \to C}\left(\beta_{B \to C}^2 + 1\right)^{1.5}\left(\beta_{A \to B}^2 \sigma_A^2 + \sigma_B^2\left(\beta_{A \to B}^2 + 1\right)\right)}{\beta_{A \to B}^2 \beta_{B \to C}^2 \sigma_A^2\left(\beta_{B \to C}^2 + 1\right) + \beta_{B \to C}^2 \sigma_B^2\left(\beta_{A \to B}^2 + 1\right)\left(\beta_{B \to C}^2 + 1\right) + \sigma_C^2\left(\beta_{A \to B}^2 + 1\right)\left(\beta_{B \to C}^2 + 1\right)^2}$$
    and $\quad \widehat{\beta}_{B_m \to A_m} = \dfrac{\beta_{A \to B} \sigma_A^2\sqrt{\beta_{A \to B}^2 + 1}}{\beta_{A \to B}^2 \sigma_A^2 + \sigma_B^2\left(\beta_{A \to B}^2 + 1\right)}$

We obtain the following probabilities by Monte Carlo approximation, resampling the 5 model parameters $100,000$ times:

Table 3: Chain orientation results in the population limit.

| Weight distribution | Chain orientation rule cases | |
|---|---|---|
| $\mathrm{Unif}((-2, .5) \cup (.5, 2))$ | $P\left[|\widehat{\beta}_{A\to B}| < |\widehat{\beta}_{B\to C}| \text{ and } |\widehat{\beta}_{C\to B}| > |\widehat{\beta}_{B\to A}|\right]$ | 29.376% |
| | $P\left[|\widehat{\beta}_{A\to B}| > |\widehat{\beta}_{B\to C}| \text{ and } |\widehat{\beta}_{C\to B}| < |\widehat{\beta}_{B\to A}|\right]$ | 5.486% |
| | $P\left[\text{"orientation rule correct on raw data"}\right]$ | **61.945%** |
| | $P\left[|\widehat{\beta}_{A_s\to B_s}| < |\widehat{\beta}_{B_s\to C_s}| \text{ and } |\widehat{\beta}_{C_s\to B_s}| > |\widehat{\beta}_{B_s\to A_s}|\right]$ | 73.181% |
| | $P\left[|\widehat{\beta}_{A_s\to B_s}| > |\widehat{\beta}_{B_s\to C_s}| \text{ and } |\widehat{\beta}_{C_s\to B_s}| < |\widehat{\beta}_{B_s\to A_s}|\right]$ | 26.819% |
| | $P\left[\text{"orientation rule correct on standardized data"}\right]$ | **73.181%** |
| | $P\left[|\widehat{\beta}_{A_m\to B_m}| < |\widehat{\beta}_{B_m\to C_m}| \text{ and } |\widehat{\beta}_{C_m\to B_m}| > |\widehat{\beta}_{B_m\to A_m}|\right]$ | 31.631% |
| | $P\left[|\widehat{\beta}_{A_m\to B_m}| > |\widehat{\beta}_{B_m\to C_m}| \text{ and } |\widehat{\beta}_{C_m\to B_m}| < |\widehat{\beta}_{B_m\to A_m}|\right]$ | 17.318% |
| | $P\left[\text{"orientation rule correct on scale-harmonized data"}\right]$ | **57.1565%** |
| $\mathrm{Unif}((-.9, -.5) \cup (.5, .9))$ | $P\left[|\widehat{\beta}_{A\to B}| < |\widehat{\beta}_{B\to C}| \text{ and } |\widehat{\beta}_{C\to B}| > |\widehat{\beta}_{B\to A}|\right]$ | 31.033% |
| | $P\left[|\widehat{\beta}_{A\to B}| > |\widehat{\beta}_{B\to C}| \text{ and } |\widehat{\beta}_{C\to B}| < |\widehat{\beta}_{B\to A}|\right]$ | 18.124% |
| | $P\left[\text{"orientation rule correct on raw data"}\right]$ | **56.454%** |
| | $P\left[|\widehat{\beta}_{A_s\to B_s}| < |\widehat{\beta}_{B_s\to C_s}| \text{ and } |\widehat{\beta}_{C_s\to B_s}| > |\widehat{\beta}_{B_s\to A_s}|\right]$ | 62.231% |
| | $P\left[|\widehat{\beta}_{A_s\to B_s}| > |\widehat{\beta}_{B_s\to C_s}| \text{ and } |\widehat{\beta}_{C_s\to B_s}| < |\widehat{\beta}_{B_s\to A_s}|\right]$ | 37.769% |
| | $P\left[\text{"orientation rule correct on standardized data"}\right]$ | **62.231%** |
| | $P\left[|\widehat{\beta}_{A_m\to B_m}| < |\widehat{\beta}_{B_m\to C_m}| \text{ and } |\widehat{\beta}_{C_m\to B_m}| > |\widehat{\beta}_{B_m\to A_m}|\right]$ | 30.025% |
| | $P\left[|\widehat{\beta}_{A_m\to B_m}| > |\widehat{\beta}_{B_m\to C_m}| \text{ and } |\widehat{\beta}_{C_m\to B_m}| < |\widehat{\beta}_{B_m\to A_m}|\right]$ | 20.607% |
| | $P\left[\text{"orientation rule correct on scale-harmonized data"}\right]$ | **54.709%** |
| $\mathrm{Unif}((-.9, -.1) \cup (.1, .9))$ | $P\left[|\widehat{\beta}_{A\to B}| < |\widehat{\beta}_{B\to C}| \text{ and } |\widehat{\beta}_{C\to B}| > |\widehat{\beta}_{B\to A}|\right]$ | 32.480% |
| | $P\left[|\widehat{\beta}_{A\to B}| > |\widehat{\beta}_{B\to C}| \text{ and } |\widehat{\beta}_{C\to B}| < |\widehat{\beta}_{B\to A}|\right]$ | 24.012% |
| | $P\left[\text{"orientation rule correct on raw data"}\right]$ | **54.234%** |
| | $P\left[|\widehat{\beta}_{A_s\to B_s}| < |\widehat{\beta}_{B_s\to C_s}| \text{ and } |\widehat{\beta}_{C_s\to B_s}| > |\widehat{\beta}_{B_s\to A_s}|\right]$ | 55.790% |
| | $P\left[|\widehat{\beta}_{A_s\to B_s}| > |\widehat{\beta}_{B_s\to C_s}| \text{ and } |\widehat{\beta}_{C_s\to B_s}| < |\widehat{\beta}_{B_s\to A_s}|\right]$ | 44.210% |
| | $P\left[\text{"orientation rule correct on standardized data"}\right]$ | **55.790%** |
| | $P\left[|\widehat{\beta}_{A_m\to B_m}| < |\widehat{\beta}_{B_m\to C_m}| \text{ and } |\widehat{\beta}_{C_m\to B_m}| > |\widehat{\beta}_{B_m\to A_m}|\right]$ | 31.867% |
| | $P\left[|\widehat{\beta}_{A_m\to B_m}| > |\widehat{\beta}_{B_m\to C_m}| \text{ and } |\widehat{\beta}_{C_m\to B_m}| < |\widehat{\beta}_{B_m\to A_m}|\right]$ | 25.136% |
| | $P\left[\text{"orientation rule correct on scale-harmonized data"}\right]$ | **53.3655%** |

We draw edge weights independently from the uniform distribution indicated in the first column of Table 3 and noise standard-deviations $\sigma_A, \sigma_B, \sigma_C$ are drawn independently from $\mathrm{Unif}(.5, 2)$ in all cases. A 99% confidence interval for the orientation accuracy under random guessing is $(49.593\%, 50.407\%)$. The orientation rule achieves above chance accuracy in all regimes.

### F.2  Finite Sample

Given observations from $(X_1, ..., X_d)$ generated by a linear ANM with either $X_1 \to X_2 \to ... \to X_d$ or $X_d \to X_{d-1} \to ... \to X_1$, we can decide the directionality by identifying the direction in which the absolute values of the regression coefficients tend to increase. More precisely, we compare the sequences of absolute regression coefficients

"*left-to-right regression coefficients*" $|\widehat{\beta}_{X_1\to X_2}|, ..., |\widehat{\beta}_{X_{d-1}\to X_d}|$

to

"*right-to-left regression coefficients*" $|\widehat{\beta}_{X_d\to X_{d-1}}|, ..., |\widehat{\beta}_{X_2\to X_1}|$.

We infer $X_1 \to ... \to X_d$ if the former is in better agreement with an ascending sorting than the latter and infer $X_d \to ... \to X_1$ otherwise.

In the main text, we discussed the case for standardized data where the regression coefficients for any two nodes $X_i$ and $X_j$ are given as $|\operatorname{Corr}(X_i, X_j)|$. We expect the sequence of absolute regression coefficients to increase along the causal order because the correlation between consecutive nodes tends to be higher further downstream as parent nodes contribute more to a nodes marginal variance relative to its noise term.

On the raw data scale, the sequences of regression coefficients are

*"left-to-right"* $\dfrac{\sqrt{\operatorname{Var}(X_2)}}{\sqrt{\operatorname{Var}(X_1)}}|\operatorname{Corr}(X_1, X_2)|, ..., \dfrac{\sqrt{\operatorname{Var}(X_d)}}{\sqrt{\operatorname{Var}(X_{d-1})}}|\operatorname{Corr}(X_{d-1}, X_d)|$    and

*"right-to-left"* $\dfrac{\sqrt{\operatorname{Var}(X_{d-1})}}{\sqrt{\operatorname{Var}(X_d)}}|\operatorname{Corr}(X_{d-1}, X_d)|, ..., \dfrac{\sqrt{\operatorname{Var}(X_1)}}{\sqrt{\operatorname{Var}(X_2)}}|\operatorname{Corr}(X_1, X_2)|.$

On both raw and standardized data, we find that the direction in which absolute regression coefficients tend to increase most corresponds to the causal direction in more than 50% of cases. To quantify "increasingness" of sequences of absolute regression coefficients we count the number of correctly ordered pairs of regression coefficients, that is, how often a regression coefficient is smaller in magnitude than regression coefficients later in the sequence and substract the number of discordant pairs. The decision rule then predicts the direction in which the sequence of regression coefficients is more increasing according to this criterion.

We apply this orientation rule to simulated data (sample size 1000) for varying chain lengths and edge distributions, and when applied to raw observational data, standardized observational data, and data when the parameters were scale-harmonized as per Mooij et al. [2020]. The table below establishes, that for iid distributed parameters of the underlying data generating process, the orientation of a causal chain can be identified with probability strictly greater than 50%.

Table 4: Empirical Chain Orientation Results

| d | edge range | accuracy by variance-sorting | | | accuracy by coefficient-sorting | | |
|---|---|---|---|---|---|---|---|
| | | raw | standardized | harmonized | raw | standardized | harmonized |
| 3 | $\pm(0.5, 2.0)$ | 97.50% | 50.05% | 84.70% | 62.58% | 73.03% | 57.30% |
| | $\pm(0.5, 0.9)$ | 80.38% | 50.05% | 69.62% | 57.15% | 62.38% | 55.65% |
| | $\pm(0.1, 0.9)$ | 65.65% | 50.30% | 60.08% | 54.17% | 55.88% | 53.45% |
| 5 | $\pm(0.5, 2.0)$ | 98.67% | 50.15% | 82.17% | 78.60% | 86.58% | 64.20% |
| | $\pm(0.5, 0.9)$ | 77.65% | 49.27% | 66.30% | 61.83% | 68.65% | 57.50% |
| | $\pm(0.1, 0.9)$ | 63.08% | 50.38% | 57.65% | 58.17% | 57.33% | 56.35% |
| 10 | $\pm(0.5, 2.0)$ | 99.38% | 50.02% | 79.30% | 93.72% | 96.97% | 69.08% |
| | $\pm(0.5, 0.9)$ | 73.75% | 50.25% | 62.00% | 64.97% | 70.70% | 58.50% |
| | $\pm(0.1, 0.9)$ | 62.55% | 51.23% | 58.25% | 55.85% | 56.05% | 54.40% |

A 99% confidence interval for the orientation accuracy under random guessing is $(47.975\%, 52.025\%)$ (1000 repetitions for each of the four noise types). Thus, variance-sorting on the standardized data is the only setting in which no above-chance orientation accuracy is achieved. This is expected, as variance sorting amounts to a random sorting once nodes are standardized.

# G   Empirical Evaluation of Varsortability

We empirically estimate expected varsortability for our experimental set-up and a non-linear version of our experimental set-up by calculating the fraction of directed paths that are correctly sorted by marginal variance in the randomly sampled ANMs.

## G.1   Varsortability in Linear Additive Noise Models

Consistent with our theoretical results, varsortability is close to 1 across all graph and noise types in our experimental set-up, cf. Table 5. Varsortability is higher in denser than in sparser graphs.

Table 5: Empirical varsortability in our experimental linear ANM set-up. Average varsortability is high in all settings. Our parameter choices are common in the literature. We sample 1000 observations of ten 50-node graphs for each combination of graph and noise type.

| | | varsortability | | |
| | | min | mean | max |
| graph | noise | | | |
|---|---|---|---|---|
| ER-1 | Gauss-EV | 0.94 | 0.97 | 0.99 |
| | exponential | 0.94 | 0.97 | 0.99 |
| | gumbel | 0.94 | 0.97 | 1.00 |
| ER-2 | Gauss-EV | 0.97 | 0.99 | 1.00 |
| | exponential | 0.97 | 0.99 | 1.00 |
| | gumbel | 0.98 | 0.99 | 0.99 |
| ER-4 | Gauss-EV | 0.98 | 0.99 | 0.99 |
| | exponential | 0.98 | 0.99 | 0.99 |
| | gumbel | 0.98 | 0.99 | 0.99 |
| SF-4 | Gauss-EV | 0.98 | 1.00 | 1.00 |
| | exponential | 0.98 | 1.00 | 1.00 |
| | gumbel | 0.98 | 1.00 | 1.00 |

## G.2 Varsortability in Non-Linear Additive Noise Models

Table 6 shows varsortabilities for a non-linear version of our experimental set-up as used by Zheng et al. [2020]. While the fluctuations in Table 6 are greater than in Table 5, all settings exhibit high varsortability on average. Our findings indicate that varsortability is a concern for linear and non-linear ANMs.

Table 6: Empirical varsortability in non-linear ANM. Average varsortability is high in all settings. Our parameter choices are common in the literature. We sample 1000 observations of ten 20-node graphs for each combination of graph and ANM-type.

| | | varsortability | | |
| | | min | mean | max |
| graph | ANM-type | | | |
|---|---|---|---|---|
| ER-1 | Additive GP | 0.81 | 0.91 | 1.00 |
| | GP | 0.72 | 0.86 | 0.96 |
| | MLP | 0.55 | 0.79 | 0.96 |
| | Multi Index Model | 0.62 | 0.82 | 1.00 |
| ER-2 | Additive GP | 0.79 | 0.91 | 0.98 |
| | GP | 0.82 | 0.89 | 0.97 |
| | MLP | 0.46 | 0.71 | 0.87 |
| | Multi Index Model | 0.65 | 0.79 | 0.89 |
| ER-4 | Additive GP | 0.90 | 0.95 | 0.98 |
| | GP | 0.74 | 0.88 | 0.93 |
| | MLP | 0.59 | 0.72 | 0.85 |
| | Multi Index Model | 0.57 | 0.73 | 0.85 |
| SF-4 | Additive GP | 0.95 | 0.97 | 0.99 |
| | GP | 0.88 | 0.94 | 0.97 |
| | MLP | 0.75 | 0.83 | 0.93 |
| | Multi Index Model | 0.77 | 0.84 | 0.97 |

## G.3 Causal Order and Marginal Variance

We observe strong empirical evidence in Figure 2 that marginal variance tends to increase quickly along the causal order, even if the settings are not guaranteed to yield high expected varsortability between a pair of root cause and effect (for example, if all edges are chosen in a small-magnitude range). This indicates that high levels of varsortability can scarcely be avoided on larger graphs.

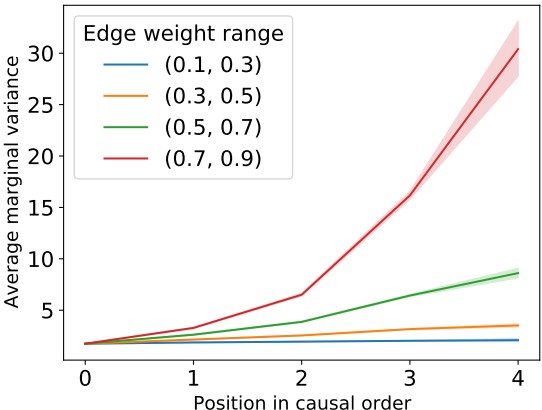

Figure 2: Average marginal variance along the causal order for 1000 observations of 1000 simulated 30-node ER-2 graphs with Gaussian noise standard deviations sampled uniformly in $(0.5, 2)$ for each edge weight range. Edge weights are drawn independently and uniformly from the union of negative and positive of the indicated edge range, that is, for example, the edge weights for the red curve are drawn from $\mathrm{Unif}((-.9, -.7) \cup (.7, .9))$.

### G.4 Varsortability Algorithm

The implementation is also available at https://github.com/Scriddie/Varsortability.

```python
import numpy as np

def varsortability(X, W, tol=1e-9):
    """ Takes n x d data and a d x d adjaceny matrix,
    where the i,j-th entry corresponds to the edge weight for i->j,
    and returns a value indicating how well the variance order
    reflects the causal order. """
    E = W != 0
    Ek = E.copy()
    var = np.var(X, axis=0, keepdims=True)

    n_paths = 0
    n_correctly_ordered_paths = 0

    for _ in range(E.shape[0] - 1):
        n_paths += Ek.sum()
        n_correctly_ordered_paths += (Ek * var / var.T > 1 + tol).sum()
        n_correctly_ordered_paths += 1/2*(
            (Ek * var / var.T <= 1 + tol) *
            (Ek * var / var.T >  1 - tol)).sum()
        Ek = Ek.dot(E)

    return n_correctly_ordered_paths / n_paths

if __name__ == "__main__":
    W = np.array([[0, 1, 0], [0, 0, 2], [0, 0, 0]])
    X = np.random.randn(1000, 3).dot(np.linalg.inv(np.eye(3) - W))
    print("Varsortability:", varsortability(X, W))

    X_std = (X - np.mean(X, axis=0))/np.std(X, axis=0)
    print("Varsortability standardized:", varsortability(X_std, W))
```

# H  *sortnregress*: A Diagnostic Tool to Reveal Varsortability

In Section 3.5 we introduce *sortnregress* as a simple baseline method. In the following subsections, we provide Python code that implements *sortnregress* thereby establishing its ease and illustrate how its DAG recovery performance reflects varying degrees of varsortability.

## H.1   Implementation of Sortnregress

The implementation is also available at https://github.com/Scriddie/Varsortability.

```python
import numpy as np
from sklearn.linear_model import LinearRegression, LassoLarsIC

def sortnregress(X):
    """ Take n x d data, order nodes by marginal variance and
    regresses each node onto those with lower variance, using
    edge coefficients as structure estimates. """
    LR = LinearRegression()
    LL = LassoLarsIC(criterion='bic')

    d = X.shape[1]
    W = np.zeros((d, d))
    increasing = np.argsort(np.var(X, axis=0))

    for k in range(1, d):
        covariates = increasing[:k]
        target = increasing[k]

        LR.fit(X[:, covariates], X[:, target].ravel())
        weight = np.abs(LR.coef_)
        LL.fit(X[:, covariates] * weight, X[:, target].ravel())
        W[covariates, target] = LL.coef_ * weight

    return W
```

## H.2   Varsortabiltiy and Score Attainable by Variance Ordering

In Figure 3 we observe that *sortnregress* improves linearly with varsortability. For a varsortability of 0.93 as in our experimental settings (cf. Section 3.3), it recovers the structure near-perfectly. *randomregress* uses a random ordering but is otherwise identical to *sortnregress*. The different ranges of varsortability can be classified as follows (n=30):

- $< 0.33$: *sortnregress* performs significantly worse than *randomregress* ($p < 1e-4$)

- 0.33–0.66: no significant difference between *sortnregress* and *randomregress* ($p = 0.40$)

- $> 0.66$: *sortnregress* performs significantly better than *randomregress* ($p < 1e-4$)

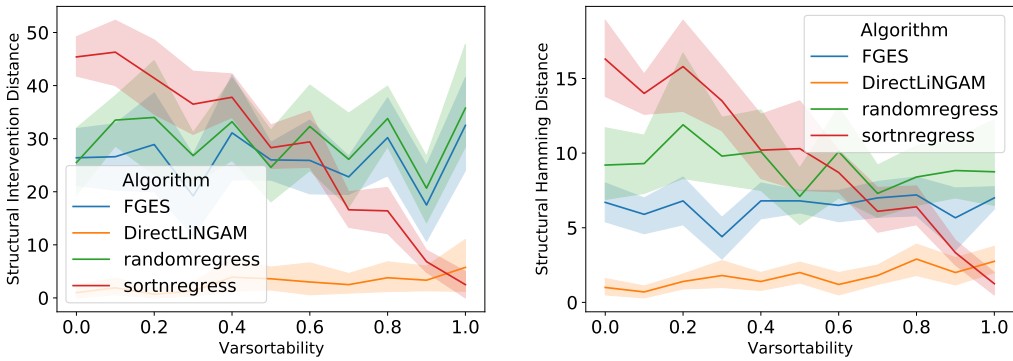

Figure 3: Relationship between varsortability and score attainable through ordering by variance. Results shown for 10 simulated 10-node ER-1 graphs in each of 10 equally spaced varsortability bins. Note that for standard simulation settings most models have high varsortability. We use edge weights in $(-0.5, -0.1) \cup (0.1, 0.5)$, Gumbel noise with standard deviations in $(0.5, 2)$, and still need to discard many models with high varsortability to obtain 10 instances per varsortability bin.

# I   Evaluation on Real-World Data

We analyze a dataset on protein signaling networks obtained by Sachs et al. [2005]. We evaluate our algorithms on ten bootstrap samples of the observational part of the dataset consisting of 853 observations, 11 nodes, and 17 edges. Our results show that there is no dominating algorithm. On average, most algorithms achieve performances similar to those of *randomregress* or the empty graph. Note that the results in terms of SHD are susceptible to thresholding choices and the empty graph baseline outperforms a majority of the algorithms. Our results are in line with previous reports [Lachapelle et al., 2019, Ng et al., 2020]. We observe scale-sensitivity of the continuous learning algorithms and *sortnregress*. However, in contrast to our simulation study in Section 4, the effect is small and inconsistent. The results do not show the patterns observed under high varsortability, which is consistent with the measured mean varsortability of $0.57$ with a standard deviation of $0.01$ across our bootstrapped samples.

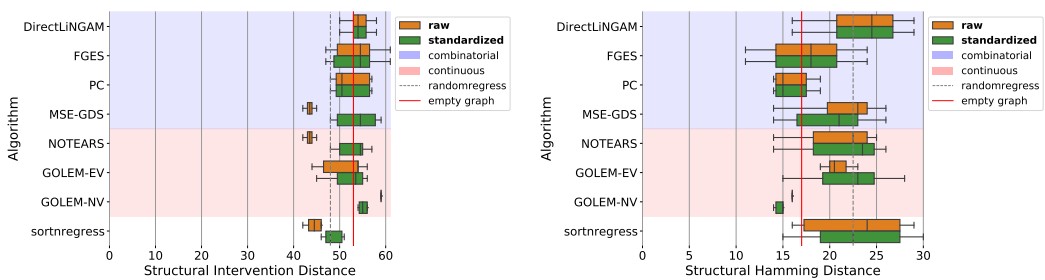

Figure 4: SID (left) and SHD (right) performance of combinatorial and continuous methods on real-world data.

# J   Model Selection in Continuous Optimization

We illustrate the optimization landscape for the Gaussian MLE under Gaussian noise. This corresponds to the loss of *GOLEM-NV* as stated in Appendix D with a sparsity penalty of zero. We compare vanilla MLE to MLE with Lasso regularization for raw and standardized data. In Figure 5 we show the loss landscape in terms of SID and SHD difference to the true structure and highlight global optima. In the case of tied scores between the true structure and an alternative structure we select the true structure. For MLE with Lasso regularization using a penalty of $0.1$, the optimal loss is achieved by the true structure more frequently under standardization (red dots accumulate in the

bottom left corner). Our result indicates that the Lasso sparsity penalty is influenced by the data scale and is better calibrated on standardized data. It is not unexpected that penalization is scale dependent, a problem that is, for example, discussed in applications of Ridge regression.

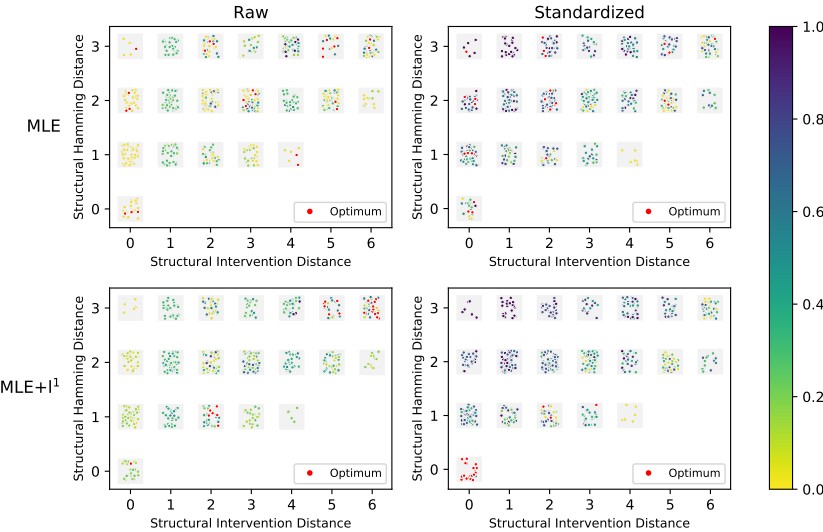

Figure 5: Standardized loss landscape for all 25 candidate graphs relative to each of the 25 possible 3-node ground-truth structures (a total of $25 \times 25$ candidate-true graph pairs). The loss is scaled to $[0, 1]$, see colorbar.

## K   Detailed Results

We provide a comprehensive overview over our empirical DAG/MEC recovery results for different evaluation metrics, graph types, and graph sizes.

### K.1   MEC Recovery

An analysis of MEC recovery allows us to distinguish whether any drops in performance are within the expectations of identifiability. We evaluate the discovery of the MEC of the ground-truth DAG in a Gaussian setting with non-equal noise variances where only the ground-truth MEC but not the ground-truth DAG are identifiable. Since evaluating the SID between Markov equivalence classes is computationally expensive and prohibitively so for large graphs, we restrict ourselves to the setting here. When comparing MEC, we choose the upper limit of SID differences in Figure 1 in the main text. In Figure 6 we show that the relative performances are similar for the lower SID limit.

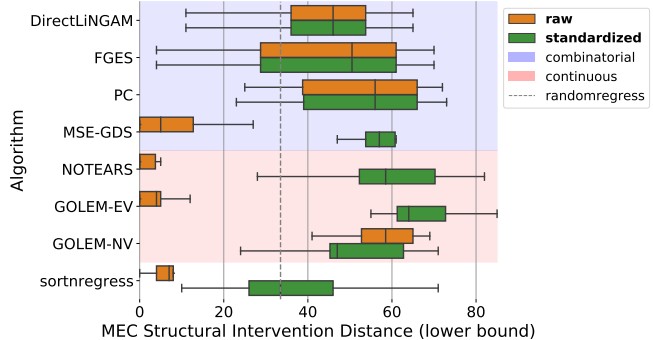

Figure 6: Lower bound of SID in MEC recovery for 10 node ER-2 graphs with non-equal Gaussian noise.

We conclude that the drop in performance extends from the recovery of the DAG to the recovery of the MEC and therefore goes beyond the difficulty of identifying the correct DAG within a MEC.

## K.2  Results Across Thresholding Regimes

To ensure the effects we observe constitute a general phenomenon, we evaluate algorithm performance for different thresholding regimes. This is especially critical on standardized data. By re-scaling the data, standardization may impact the correct edge weights between nodes, potentially pushing them outside the thresholding range. Following Zheng et al. [2018], Ng et al. [2020], we perform thresholding for the continuous structure learning algorithms and prune edges with an edge weight in the recovered adjacency matrix of less than 0.3. If the returned graph is not acyclic, we iteratively remove the edge with the smallest magnitude weight until all cycles are broken. We find that the qualitative performance differences between raw and standardized data are robust to a wide range of threshold choices.

Figure 7a and Figure 7b show SID performance for different thresholds. Even though the thresholds are orders of magnitude apart, a comparison reveals that the relative performances are nearly identical.

We observe that SHD performance is also robust across different thresholding regimes. Figure 7c shows performance using *favorable* thresholding. In this regime, the threshold leading to the most favorable SHD performance is applied to each instance individually. Figure 7d shows performance for a fixed threshold of 0.3. A comparison reveals nearly identical relative performances in both cases.

Overall, we observe that the effect of varsortability is present even for the most favorable threshold in case of SHD, and for a wide range of thresholds in case of SID, where computation of a favorable threshold is computationally infeasible.

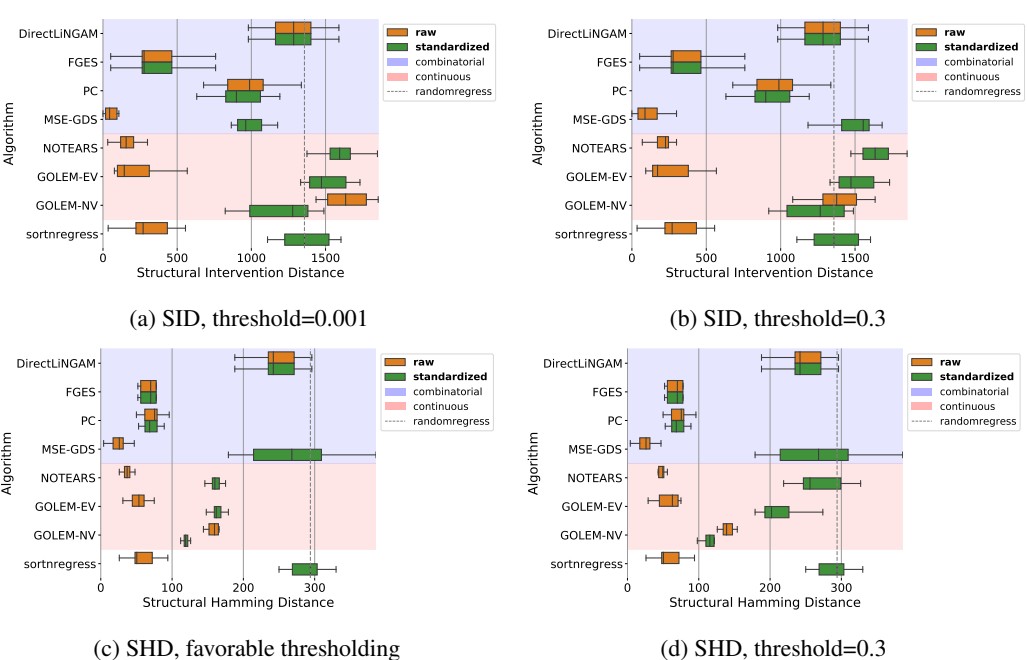

(a) SID, threshold=0.001      (b) SID, threshold=0.3

(c) SHD, favorable thresholding      (d) SHD, threshold=0.3

Figure 7: Results for different thresholding regimes. Gaussian-NV noise, ER-2 graph, 50 nodes.

## K.3  Results Across Noise Distributions and Graph Types

Figure 8 and Figure 9 show algorithm comparisons in terms of SID and SHD, respectively. The differences in performance on raw versus standardized data are qualitatively similar regardless of the noise distribution. We showcase results for different graph types in the non-Gaussian setting. *DirectLiNGAM* performs well only in the non-Gaussian cases, as is expected based on its underlying identifiability assumptions.

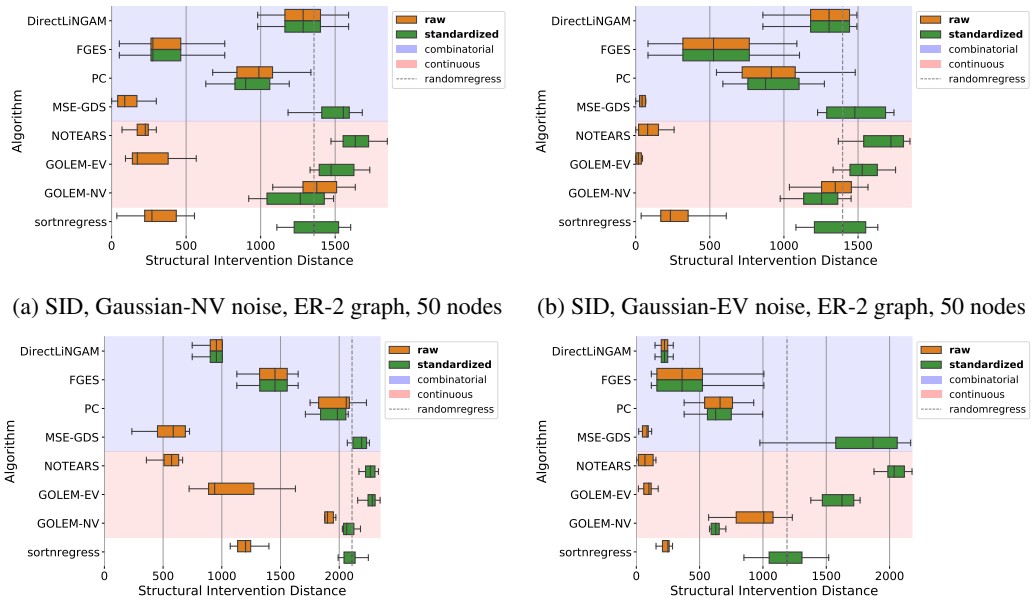

(a) SID, Gaussian-NV noise, ER-2 graph, 50 nodes

(b) SID, Gaussian-EV noise, ER-2 graph, 50 nodes

(c) SID, Exponential noise, ER-4 graph, 50 nodes

(d) SID, Gumbel noise, SF-4 graph, 50 nodes

Figure 8: SID results across noise types and for different graph types with 50 nodes

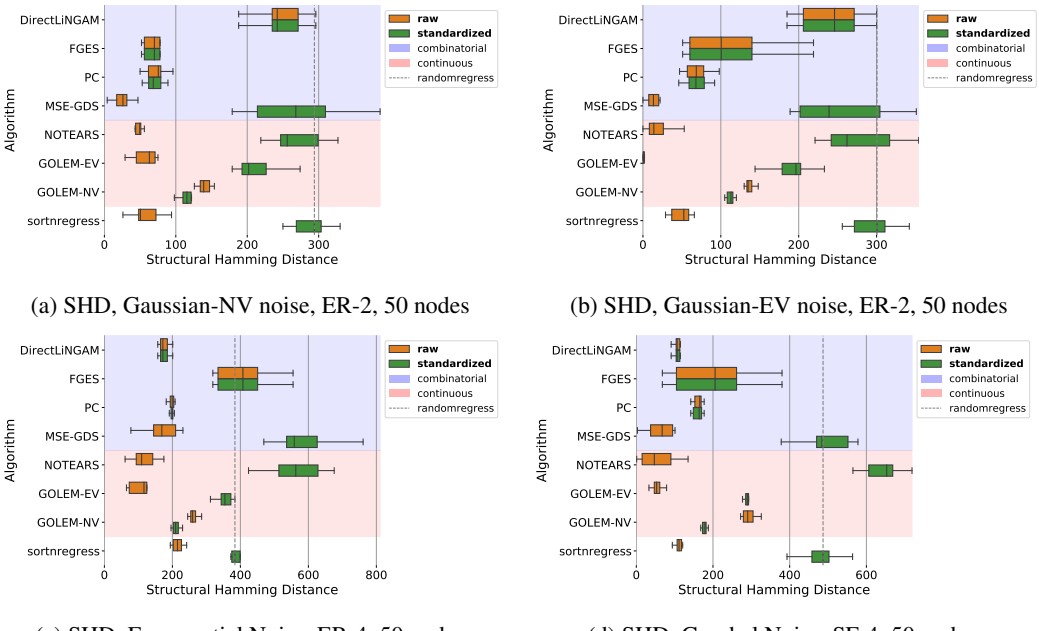

(a) SHD, Gaussian-NV noise, ER-2, 50 nodes

(b) SHD, Gaussian-EV noise, ER-2, 50 nodes

(c) SHD, Exponential Noise, ER-4, 50 nodes

(d) SHD, Gumbel Noise, SF-4, 50 nodes

Figure 9: SHD results across noise types and for different graph types with 50 nodes

## K.4 Results Across Noise Distributions, Graph Types, and Graph Sizes

The following experimental results largely follow earlier settings and results by Zheng et al. [2018], Ng et al. [2020].

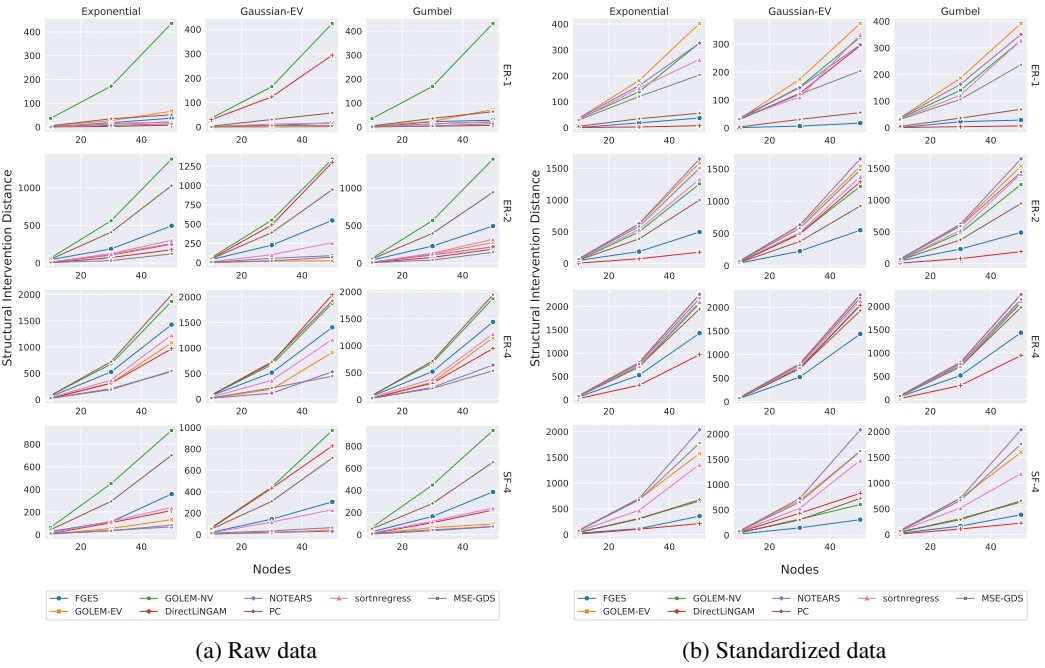

(a) Raw data

(b) Standardized data

Figure 10: SID results across noise types, graph types, and graph sizes.

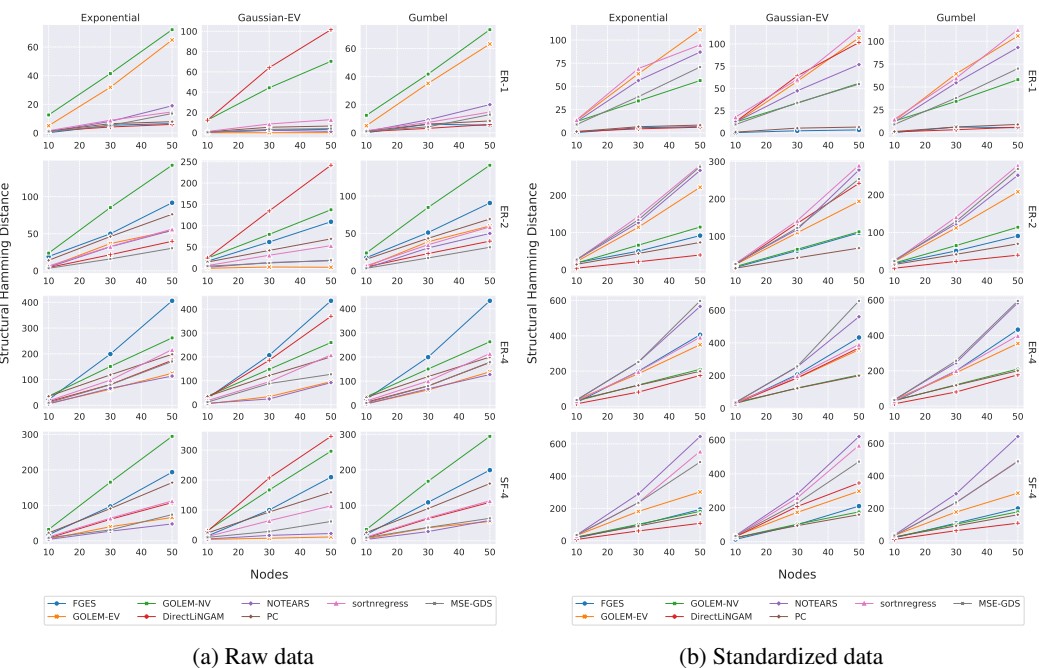

(a) Raw data

(b) Standardized data

Figure 11: SHD results across noise types, graph types, and graph sizes.