# OpenReview forum: "Beware of the Simulated DAG! Causal Discovery Benchmarks May Be Easy to Game"
_NeurIPS.cc/2021/Conference — NeurIPS 2021 Poster_

### Official Review · Reviewer_b4Yx · 2021-07-14

**Rating:** 7
**Confidence:** 4

**Summary:**

The paper considers common DAG simulation schemes and and defines and studies a property that results, which does occur in real data, but which may be specifically exploited by continuous causal structure learning algorithms like NOTEARS and GOLEM which case causal structure learning as a differentiable optimization problem, essentially allowing them to game common benchmarks.

The key property that results from common DAG simulation schemes is downstream nodes in the causal graph having higher variance. The paper defines "varsortability" as the fraction of directed paths of any length which start from a node with strictly lower variance than the node they end in. The paper finds that forward sampling in the linear and additive noise model settings result in varsortability of ~ .7 and ~.9, respectively.

The paper then goes on to show how varsortability guides the gradient-based steps when continuous causal structure learning algorithms like NOTEARS and GOLEM are used.

Next, the paper caries out simulations which compare combinatorial vs. continuous causal structure learning algorithms (and a baseline that relies only on varsortability to obtain the causal order) in settings where the simulated data is standardized and not standardized. The results show that continuous causal structure learning algorithms show significantly higher performance due to varsortability whereas combinatorial algorithms are roughly unaffected.

The paper further elaborates that even standardized data where prior to standardization there was high varsortability may still allow allow gaming due to distinct covariance patterns.

**Limitations And Societal Impact:**

Not really discussed - but I don't foresee negative impacts

**Main Review:**

The paper presents a very clear and thoughtful analysis of current methods for benchmarking causal discovery algorithms and identifies an important problem, which I don't think has previously been addressed rigorously.

Since the paper also addresses the core question of how we should be benchmarking causal discovery algorithms I think the significance and novelty are both high.

Finally the paper is well written and organized, reviews existing literature thoroughly and the main observations regarding varsortability and continuous causal discovery algorithms are well supported with experiments.

I don't have any significant concerns. My main criticism would be the claims in section 5 remain unclear - this may be more clear in the appendix (which I am unable to review given the overload of papers this NeurIPS cycle), but I still support accepting this paper based on the content in the other sections.

**Time Spent Reviewing:**

3

---

> ### Author Response · Authors · 2021-08-10
> **We add a nonlinear greedy MSE-based search as further evidence to section 5.2**
>
> We thank you for highlighting the significance and novelty of our contribution for causal discovery benchmarking.
> We indeed hope that appendix D clarifies the claims in Section 5.1.
> To further substantiate our points in Section 5.2 on varsortability in nonlinear settings, we provide a new empirical analysis.
> We implemented a nonlinear greedy DAG search (GDS) algorithm based on off-the-shelf sklearn NuSVR which inserts the edge that reduces the MSE the most in a stepwise forward manner.
> We find that it outperforms GraN-DAG in the nonlinear ANM settings presented in Table 2, which corroborates the hypothesis that varsortability may be similarly important to structure learning performance in nonlinear settings as it is in linear settings.
>
> |graph|SID GDS|SID GraN-DAG|
> |---|---|---|
> |ER-4 |66.5 +- 39.28|165.1 +- 21.0|
> |SF-4 |36.3 +- 18.15|62.5+- 18.8|
>
> Runtime of the greedy search is in the order of 2 minutes for 50 node graphs on a regular laptop and the implementation will be provided in the accompanying code repository in addition to sortnregress and varsortability.

---

### Official Review · Reviewer_ciTV · 2021-07-16

**Rating:** 7
**Confidence:** 3

**Summary:**

This paper shows that one needs to be cautious when rescaling your data prior to using a causal structure learning algorithms. The authors introduce the concept of varsortability that measures the agreement in how much the marginal variance tends to increase along the causal order. They show that standardization of your data can hurt performance in identifying the DAG (or its equivalence class) which can be explained by varsortability. The authors claim that this concept of varsortability also explains why even after standardization certain continuous structure learning algorithm perform well. The authors focuses on additive noise models and perform an extensive benchmark.


**Limitations And Societal Impact:**

One of the limitations of this work was already mentioned in the main review section. Moreover, the authors discuss the further limitations, such as examining the role of varsortability in non-linear and empirical settings in detail in Section 6. The authors aim to raise awareness of the severity by which scaling properties in data may distort structure learning algorithm performance makes this paper a valuable contribution to NeurIPS.


**Main Review:**

The paper is very well written and a pleasure to read. The authors shed some new light on current causal structure learning algorithms by introducing the intuitive concept of varsortability. They show that varsortability is high in common simulation schemes used for benchmarking causal structure learning algorithms. Interestingly, they show that high varsortability may still be exploited after data standardization and may explain why certain continuous structure learning algorithms work so well. They motivate this possible explanation by matching the performance of their algorithm sortnregres to other algorithms. These results, however, do not imply, or proof, that varsortability is a "causal" explanation of why other continuous learning algorithms work so well in the presence of high varsortability. In Section 3.4 the authors give a heuristic argument why this is the case, but the paper would be much stronger if the authors could actually proof this, and state precisely under which circumstances this would hold. Part of the arguments of this section could I think be improved upon. For example, the authors state that:
- line 239: "We expect the residuals after the first gradient step ... to be larger ..."
- line 254: "b) why performance changes once data is standardized ... causal order."
without further explanation. It would be helpful if the authors could elaborate more on this. Nevertheless, the concept of varsortability and the benchmarking results will still be a valuable contribution to NeurIPS.

Minor comments:
- line 217-221 "In general, ... causal order.": What motivates these claims? I think a reference would be appropriate.
- line 226-229: "We do not expect combinatorial ... to be dependent on data scale."  Why is that?
- line 310 "are driven by": Really? Or should this read as "may be driven by"?
- line 292 "sortnregress": Why not emphasize more that this is a novel contribution?
- Why Figure 1 only for ER-2 graphs and Table 2 only for ER-4 and SF-4 graphs?


**Time Spent Reviewing:**

5h

---

> ### Author Response · Authors · 2021-08-10
> **We add a variance-sensitive combinatorial algorithm and incorporate the suggested limitation.**
>
> Thanks a lot for your thorough feedback and for highlighting the value of our contribution for causal discovery benchmarking purposes.
> We agree that future research into the optimization for continuous structure learning would be desirable to help understand how precisely their results are impacted by varsortability.
> To provide further evidence for the claim that their performance is largely due to the score function exploiting varsortability, we have added a new set of empirical results.
> We find that a greedy DAG search (GDS) using the same MSE score as NOTEARS performs as good as or better than NOTEARS on raw data while suffering from the same drop in performance for standardized data in the settings shown in the main text.
> The results (cf. Figure 1) on 10 raw instances of 50 node ER-2 graphs compare as follows:
>
>   |SID results |min |q25|median|q75|max|
>   |---|----|--------|-----|------|-----|
>   |NOTEARS|32.0| 162.75 |221.5 |245.25|367.0|
>   |GDS|0.0 | 27.75| 50.5 |119.75|300.0|
>
>
>
>   |SHD results |min |q25|median|q75|max|
>   |---|----|--------|-----|------|-----|
>   |NOTEARS|19.0|42.00| 48.5 |51.50 | 70.0|
>   |GDS    |1.0 | 9.75| 22.5 |30.50 | 51.0|
>
> Moreover, we have added a nonlinear version of the greedy DAG search and find that it outperforms the results of GraN-DAG (Lachapelle et al. 2019) reported in Table 2.
> We will provide the implementations of linear and nonlinear greedy DAG search with our codebase upon publication.
>
> We are aware that this cannot prove that continuous algorithms excell for the same reasons, but hope it can corroborate the point that variance-sensitive score functions may be a main driver of excellent performance on varsortable raw data and the observed performance drop upon standardizing the data.
> Furthermore, it emphasizes the simplification of the structure learning problem if varsortability is high and shows that even simple algorithms can obtain state-of-the-art results in these settings.
>
> Regarding the examples, we will incorporate your suggestions as follow:
>
> 1. Line 238/239. So far, our statement is based on empirical findings and we have not provided a rigorous formal argument. We observe a strong positive correlation between the ordering by variance and the ordering by residuals after the first step. For small learning rates, the correlation is perfect during the first optimisation steps. (For NOTEARS' MSE-based loss function, we expect that the key is that due to continuity of the residuals X-X(a(X'X)) after the first gradient step w.r.t. the step size a, there always exists a sufficiently small positive step size such that the order of residuals after the first step agrees with the marginal variances. Future work may investigate arguments for the more subtle case of GOLEM-NV/-EV and subsequent optimisation steps.)
>
> 2. Line 254. We will rearrange the sentences to clarify that this is a conjecture and add the open conjecture to the limitations in Section 6 for future research.
>
> Regarding your minor comments, we will address them in the following manner to improve our presentation:
>
> 1. Line 217-221. We will change the statement as follows:
> _"Motivated by the strong impact of varsortability on some structure learning algorithms (cf. Appendix F.2), we advocate an empirical evaluation and reporting of varsortability when simulating additive noise models (cf. Appendix E.3 the implementation)."_
>
> 2. Line 226-229. Thank you for bringing this to our attention. The statement was imprecise and the emphasis was on constraint-based/equivalent score function/residual independence test-based (combinatorial) methods. We do not expect to see the effect for combinatorial algorithms that use a score-equivalent criterion (or scale-independent (conditional) independence tests) and will include this in our revised statement. To further clarify this distinction in the article, we now include a combinatorial method that is affected by data scale: MSE-based GDS.
>
> 3. Line 310. We agree and will change the statement to _"may be driven primarily by high varsortability."_
>
> 4. Line 292. Thanks a lot for the suggestion. We will rephrase the statement as follows:
> _"We propose a novel baseline algorithm termed sortnregress which will serve as a reference marking the performance achievable by directly exploiting marginal variances."_
>
> 5. We choose the settings which are predominantly used in the respective original papers of NOTEARS, GOLEM, and GraN-DAG. Due to space constraints, further results are shown in the Appendix.
>
> We will add the open conjecture and the limitation you raise about a "causal explanation of performance drop" to our limitations section.

---

### Official Review · Reviewer_BrkD · 2021-07-21

**Rating:** 6
**Confidence:** 3

**Summary:**

This submission is about causal structure learning and how some simulated directed acyclic graphs (that are used for benchmarking structure learning approaches, and that are generally used for causal representations) can have properties (in particular, marginal variances and also covariance patterns) that could render some structure learning algorithms more efficient, even though that might be not necessarily replicable for real-world data; that is highlighted a concern.

The authors describe a measure, varsortability, that relates to a relationship between marginal variances and the causal structure in a causal graph. They discuss how identifiability might be affected by that. The submission shows how, and explains potentially why, the varsortability of causal graphs and the data simulated/generated from them might interplay with the performance of some structure learning algorithms (in particular, continuous structure learning methods), and how that performance might be explained and depend (potentially, inadvertently) up to some significant extent by the varsortability (in particular, for simulated DAGs). The submission covers that through some theoretical analysis and empirical studies. Because marginal variances and varsortability depend on the choice of the scales of variables, the submission raises concerns about the usefulness/interpretation of existing benchmarks (particularly if no experiments with data standardisation are performed) and their results for existing structure learning algorithms. The submission also presents a baseline structure learning method that relies on marginal variances; that method can be used as a baseline for benchmarking. The theoretical results in the submission is for linear additive noise models; the empirical experiments cover both some linear and non-linear settings. The empirical experiments (and results) cover simulated and also some real-world data. In addition to discussing the points that relate to marginal variances, the submission also has a part that relates to covariance patterns and how those might influence the benchmarking in a similar way. The paper finishes with a conclusion and discussion, including some recommendations/comments as part of that.


**Limitations And Societal Impact:**

* There is a question: to what extent is high varsortability natural in the real world? Is it generally natural? (E.g. for variables where scales can be chosen.) Authors acknowledge this question too (e.g. in lines 375-378). I wonder whether it should be covered a bit more earlier in the paper as well (e.g. in the abstract or on the first pages)?
** If that is the case (that it is natural), then would it be appropriate to use algorithms that rely on scales in practice (as far as scales are (somewhat) consistent)? Even though (some of) their performance is explained by varsortability, is it fine (and desired?)?

* (I assume it is a general issue in the field, and that is not trivial to resolve:) There seems to be a limited set of real-world, expert-annotated datasets.
** It is also interesting to note, based on the authors’ note about that, that on the one real-world dataset that was used in the paper, for SID the algorithms perform generally quite similarly (the appendix, line 761) and, if I understand correctly, not very well (although NOTEARS on the raw data seems to be performing relatively well on SID, and sortnregress too).

*

Societal impact:

There are generally potential huge positive impacts of the paper, and that is really appreciated: the submission raises questions and contains work that are important for the field and its applications (and that is expected to lead to positive social impact). The authors make a statement regarding that in the paper checklist.

(Further study is required to understand better e.g. whether high varsortability is natural for some real-world data (given some assumptions about scale relationships); the authors make a similar statement in lines 375-378 regarding additive noise models. Note that those further studies would affect further evaluation of different (existing and new) structure learning algorithms (and development of new ones), and further analysis and development of benchmarks and metrics. An obvious potential negative social impact might be if e.g. (a hypothetical scenario) varsortability of real-world datasets would be for some reason generally originally underestimated and if that would for some reason lead to bias towards some algorithms (e.g. if they don’t perform well on standardised data) until later it would be established that high varsortability is actually a natural property (for some real-world data) and identifiability significantly depends on it (again, that is just a hypothetical scenario).)


**Main Review:**

The paper is written very well. I believe it is a very important paper. The paper is significant because it raises valid concerns about existing benchmarking (and results received with them) in a constructive way (see the summary section of the review for more details on the contribution). Thanks extremely much to the authors for their work on this.

Many related work papers are referred to in the submission. The submission naturally relies on previous work. Based on that coverage, it is assumed that the idea and implementation, as a whole, are original.

Some clarity points:

* The sentence on lines 53-56: it might be helpful to clarify for a reader whether this statement (“... cannot rely …”) is covered in those references or it is a finding/point of this submission.

* Line 90: “inadvertent” (also similarly in line 319 and “inadvertently” in line 1) - are they inadvertent or potentially “advertent”/natural (and the submission also refers to this question in lines 375-377)? (As far as the variables have the same meaning and it is natural for them to have the same scale?)

* Some clarification/more details might be helpful for lines 169-177:
** Line 172: “... obtaining varsortability for each individual …” - is something like “a varsortability component/part” meant? (since a pair of nodes is just a part of a graph)
** Regarding sentences “Even … quickly” and “The further … to the causal order” - I assume the authors might be (partially) relying on the Appendix B regarding those statements. However, those statements are not trivial, to the best of my understanding, and a proof (or at least a sketch) might be essential for them. (For example, can there be some cancellation of noise on a causal path along a causal order? (As the authors mention as well in lines 212-214.))
** Philosophically, how can we be certain that those properties are not replicated in the real world (for at least some data)?

* Line 179: “If varsortability v = 1, the causal structure is identifiable” - this is a strong statement. I see how it could be potentially (sometimes?) identifiable (from a “partial” causal order), but I don’t see why the causal structure would be necessarily identifiable in general, especially from noisy data. A proof, a sketch and/or a reference would be very helpful.

* Lines 193-194: if we don’t know the true data scale (and maybe there is no _true_ scale) but variables have the same meaning (e.g. seconds or meters), would a scale matter (at least for ANMs) as far as the scale that is used the same (e.g. seconds and seconds but not hours)?

* Figure 1: “The performance of _sortnregress_ … matches that of …” - it is not an exact match. (And someone could argue NOTEARS performs somewhat better than sortnregress in some cases, although it is not clear whether there are significant differences.) It might be worth highlighting that.

* Line 358: “later” - just to check, later to this submission’s first publication (e.g. on arXiv)?

Some other/minor points:

* It might be helpful to define “DAG” in the abstract.

* Line 9: “where varsortability may be absent” - can a measure be absent? (Is it meant that varsortability is low or equal to 0?)

* Line 10: “no longer identify” - do they rather just (sometimes) identify it, generally, much worse? (They still might perform somewhat better than the “randomregress”, e.g. see Figure 1, top-right. Before, they also did not identify it perfectly.)

* Lines 14-15: “[existing?] generic benchmarks”

* Line 43: it might be helpful to define the “MSE”. (It is appreciated that there is some definition after the line 139.)

* Line 94: “i.i.d.” presumably refer to any x^(i) overall (i can be any from 1 to n) and not to dimensions - it might be not clear from the first read although that is probably obvious in this context.

* Line 101: “if and only if” - a reference to a proof might be helpful, even if trivial.

* The equation after line 159: it might be helpful (although probably trivial) to describe what does “i -> j \in E^k” mean.

* It might be helpful to cover what happens with the varsortability if e.g. a graph does not have any edges.

* More derivation details for section 3.4 might be helpful (e.g. in an appendix section).

* Can varsortability be measured (approximately, using empirical variances) for the real dataset (as in the appendix G)?

* Line 328, “from finite-sample to the population setting” - just to check, is it a reference to sections “D.1 Infinite Sample” and “D.2 Finite Sample” in the appendix (or to some other section(s))?

* Line 360: “score-equivalend” -> “score-equivalent” ?

* Line 371: “is needed to .” - potentially, some missing (or extra) words?

* Note that the appendix was looked at (for some parts) but was not carefully checked.

*

Given some of the clarity points, although I think this work is very important, I am sorry, sadly I don’t feel comfortable at this point to recommend its acceptance. There is a chance though that: (a) I did misunderstand some points; (b) some points are obvious and they just were not mentioned by the authors; (c) or/and some points are valid and they can be addressed by the camera-ready version submission if the paper is accepted. Given the authors’ rebuttal, to which I am looking forward, there is a chance that the score will be adjusted. Given the importance of the paper, it probably would be very beneficial for the field if it is accepted to facilitate the discussion on this important topic.

All parts of the review are provided to the best of my understanding and knowledge. Mistakes/misunderstandings in the review are possible. I kindly ask to point me to them, if any, if possible, please.

*

* A discussion point (of the review): The paper topic relates to an important question on whether scales are important for structure learning tasks (and how should benchmarks be constructed; and what should metrics be). If some variables have the same meaning, does it make sense (and is it essential for full possible “identifiability”) to use their scales? (For example, if two variables concern time, is it natural to use seconds and seconds (without any standardisation)? And also, potentially, generally to use a consistent metric system, assuming there are some natural connections between different base units, or/and they can be naturally expressed through one another?)
** (It also brings an interesting question on how a structure learning algorithm can try to incorporate information about different units that can’t be necessarily naturally expressed through each other (e.g. numbers of people and currency (money) units) using some meaningful statistical information that describes some relationships between them (e.g. an average income or some average expense per a person); but it is out of the scope of this review.)

*

**For the update, please, see my message dated the 2nd of September 2021 (GMT).**


**Time Spent Reviewing:**

I have not tracked time. I generally conduct a review in a few sub-iterations, to be able to reflect (between them) on a submission/material under review for some time.

---

> ### Author Response · Authors · 2021-08-10
> **We address all points to improve clarity**
>
> We thank you for your thorough review and for highlighting the importance, constructiveness, and originality of our paper.
> Below, we provide a point-by-point response following the order of remarks in your review and describe how we address each point in the camera ready version.
>
> ### Point-by-Point Response
>
> 1. We will highlight that sensitivity to data scale is commented on in the references mentioned in line 53-56 and part of our contribution is to highlight its severe impact on graph recovery performance including methods not covered in those references.
>
> 2. We agree that whether or not and in what context varsortability is "natural" is an important question for future work. We will qualify "inadvertent" throughout the article and write, for example, "potentially inadvertent patterns" in line 319, to reflect the openness of the question.
>
> 3. We thank the reviewer for their suggestions on how to make 169-177 more precise.
>
>     1. Line 172. We will change the statement to _"If the weights and noise variances are drawn at random, their distributions determine whether the causal order of any two connected nodes in the graph agrees with the order of increasing marginal variance and whether the node pair thus contributes to a higher varsortability of the graph."_
>
>     2. We agree that more details are helpful. We will add a plot to supplementary E where we show how variance increases along the causal order for graphs with different varsortabilities and will update the statement in line 174-176 with the following explanation:
>     _"We observe that even for modest probabilities of any two neighboring nodes being correctly ordered by their marginal variances, the variance of connected nodes along the causal order tends to increase for many instantiations of linear and non-linear additive noise models (cf. Appendix E). This empirical finding can be explained as follows:
>     We obtain the variance of a node by adding each ancestors' noise variance multiplied with the squared product of the corresponding paths' coefficients to its own noise variance.
>     Cancellation of noise terms along different paths is possible but unlikely if edge weights are drawn independently (cf. Meek 1995 for why exact cancellation, i.e., faitfhulness violations are unlikely).
>     Consequently, the further apart a connected pair of nodes along the causal order, the greater the chance the descendant node accumulates more variance along the path in addition to the ancestor's variance."_
>
>     3. Varsortability may be replicated in the real world, as mentioned, for example, in the abstract and discussion section. Our contribution highlights that if current data simulation practices aim to reflect the real world, this implicitly amounts to assuming that there is a high degree of varsortability on the raw data scale. The philosophical burden of proof is to substantiate this assumption. We hope our article raises awareness and motivates future research into the plausibility and suitability of the varsortability assumption for real-world structure learning tasks. We also advocate reporting varsortability measures for simulated data so that one can choose the appropriate algorithm for the varsortability level at hand. To provide a first point of evidence regarding real-world varsortability, we will include the varsortability (mean=0.57, std=0.01, for our bootstrapped samples) of the real-world dataset in line 311.
>
> 4. The identifiability is given through the same procedure used in sortnregress and holds in the sample limit. Thank you for pointing this out. We will add the following sentence:
> _"This is the case because at v=1, an ordering by marginal variance is a valid causal ordering and sparse regression of each node onto its predecessors then enables reconstruction of the graph in the sample limit under mild assumptions (cf. Bühlmann et al. 2014)."_
>
> 5. Thank you for raising this question, which we address by adjusting our statement in line 193 as follows:
> _"For real-world data we cannot readily assess nor presume varsortability as we do not know the parameters and data scale of the data generating process."_
> We will also make explicit that varsortability may be leveraged if known to be present for given data scales in line 190.
>
> 6. In light of the similar results of NOTEARS, GOLEM, and sortnregress, we believe that further research is needed to delineate which settings are more amenable to which algorithm, hyperparameters, and thresholding choice.
>   To further substantiate our claims that a variance-sensitive score function is a primary driver of excellent results on varsortable data, we have conceived an additional experiment.
> We implemented a greedy forward DAG search (GDS) using a MSE score (that at each step inserts the edge that reduces the MSE the most when linearly regressing nodes onto their currently selected parents) and find that it performs on par with or better than all other algorithms on raw data using the settings presented in Figure 1 in the main text.
> On 10 raw instances of 50 node ER-2 graphs, the results for NOTEARS and GDS can be seein in the first table below. We observe a matching drop upon data standardization for both. We find similar results on nonlinear data shown in the second table below which we will add in Section 5.2. The implementation relies on sklearns NuSVR implementation without parameter tuning. For reproducibility, we will include the linear and non-linear GDS implementations in our code repository.
>   |measure| algorithm |min |q25|median|q75|max|
>   |---|---|----|--------|-----|------|-----|
>   |SID|NOTEARS|32.0| 162.75 |221.5 |245.25|367.0|
>   |SID|GDS|0.0 | 27.75| 50.5 |119.75|300.0|
>   |SHD|NOTEARS|19.0|42.00| 48.5 |51.50 | 70.0|
>   |SHD|GDS    |1.0 | 9.75| 22.5 |30.50 | 51.0|
>
>     |graph|SID GDS|SID GraN-DAG|
>     |---|---|---|
>     |ER-4 |66.5 +- 39.28|165.1 +- 21.0|
>     |SF-4 |36.3 +- 18.15|62.5+- 18.8|
>
> 7. We will clarify this in the camera-ready version as follows:
> _"Following the first release of the present paper, the drop in performance of NOTEARS upon standardizing the data was also independently reported by Kaiser and Sipos 2021 who present a low-dimensional exemplary case."_
>
> ### Varsortability in the Real World and Societal Impact
>
> Regarding your discussion point and potential limitations of our work, we do agree that varsortability may hold and thus be legitimately exploited in some settings and for some measurement scales. We note that this concerns and motivates future work and discussion but does not question the relevance, originality, or correctness of our present work. Our article contributes diagnostic tools and lays the ground for future research into varsortability as a largely overlooked dimension/property in simulated DAGs that severely affects structure learning performance. We thank you for highlighting the potential benefit and timeliness of our contribution to the community. We believe we improved upon the clarity of the points you raised.
>
> Regarding societal impact, we disagree with a potential negative societal impact of our research. On the contrary, we believe it has a positive impact as it raises awareness of a previously overlooked benchmarking determinant and fosters a better understanding of their implicit assumptions relevant for real-world applications.
>
> ### Thank You
>
> We thank you for your diligent reading and will address all minor points not explicitly responded to above in the camera-ready version.
> We are looking forward to a continued great discussion. Thanks again for the thoughtful exchange and comments!

---

> > ### Comment · Reviewer_BrkD · 2021-08-15
> > **Response**
> >
> > Dear Authors,
> >
> > Thank you very much for your response. It is very helpful.
> >
> > R1.1. “We obtain the variance of a node by adding each ancestors' noise variance multiplied with the squared product of the corresponding paths' coefficients to its own noise variance.”
> >
> > I am sorry, I might be missing something, but should potentially covariances of ancestors’ nodes be included there generally as well?
> >
> > R1.2. “Cancellation of noise terms along different paths is possible but unlikely if edge weights are drawn independently (cf. Meek 1995 for why exact cancellation, i.e., faitfhulness violations are unlikely). Consequently, the further apart a connected pair of nodes along the causal order, the greater the chance the descendant node accumulates more variance along the path in addition to the ancestor's variance.”
> >
> > I appreciate that full cancellation is unlikely, but, I am sorry, I still don’t see why variance necessarily increases along the path due to covariance terms. I do appreciate though that for some “generative models” that are used to sample synthetic causal models that might be generally the case. That is, is it fair to say that the chance is generally greater for (some of) _existing_ approaches that are used to generate synthetic models and data?
> >
> > (Also, there is potentially a typo: “faitfhulness”.)
> >
> > R1.3. “To provide a first point of evidence regarding real-world varsortability, we will include the varsortability (mean=0.57, std=0.01, for our bootstrapped samples) of the real-world dataset in line 311.”
> >
> > This is very helpful and interesting, thanks a lot for calculating that. If I understand correctly, that is an estimate of the varsortability for the real-world dataset, is this correct? Could you advise me, please, what do you mean by “bootstrapped samples”?
> >
> > I am curious though, would the mean value of 0.57 with the low standard deviation (0.01) be considered not high? As in, are the lines 310-311 still consistent with that estimate? (I am assuming it is the same dataset.)
> >
> > R1.4.
> > > The identifiability is given through the same procedure used in sortnregress and holds in the sample limit. Thank you for pointing this out. We will add the following sentence: "This is the case because at v=1, an ordering by marginal variance is a valid causal ordering and sparse regression of each node onto its predecessors then enables reconstruction of the graph in the sample limit under mild assumptions (cf. Bühlmann et al. 2014)."
> >
> > Thank you very much. I think this clarification is helpful.
> >
> > However, while intuitively it makes sense that for some models under some assumptions and in limit we could recover structure, it still would be helpful to refer to/show a general proof of this. (For example, it is generally less intuitively obvious for cases with a structure where node A is a parent for B and C; and B is a parent for C as well.)
> >
> > Could you advise me, please, when you refer to “Bühlmann et al. 2014”, do you mean “CAM: Causal additive models, high-dimensional order search and penalized regression”?
> >
> > If yes (I think so), and assuming you might be using version 2 (“1 Dec 2014”) from arXiv (arXiv:1310.1533), do you refer to section 2.5, page 11, equation (12), or/and to something else? If so, would that equation (as that paper writes: “[...] all relevant variables (i.e., edges) are selected”) cover the full statement about all cases where the varsortability equals 1 (e.g. including absences of edges in order to fully identify the causal structure)?
> >
> > R1.5.
> > > Thank you for raising this question, which we address by adjusting our statement in line 193 as follows: "For real-world data we cannot readily assess nor presume varsortability as we do not know the parameters and data scale of the data generating process."
> >
> > I am sorry, just to check and to follow up on my original review, is it a general statement, or is it a statement for some cases but not necessarily all? That is, as per my original review, e.g. if all variables have the same meaning (e.g. seconds or meters), would a scale matter (at least for ANMs) as far as the scale that is used the same (e.g. seconds and seconds but not hours)?
> >
> > R1.6.
> > >> Figure 1: “The performance of sortnregress … matches that of …” - it is not an exact match. (And someone could argue NOTEARS performs somewhat better than sortnregress in some cases, although it is not clear whether there are significant differences.) It might be worth highlighting that.
> > > In light of the similar results of NOTEARS, GOLEM, and sortnregress, we believe that further research is needed to delineate which settings are more amenable to which algorithm, hyperparameters, and thresholding choice.
> > > …
> >
> > Thank you. I am sorry, one of my original points about “The performance of sortnregress … matches that of …” was not addressed in the rebuttal, if I understand correctly. A comment about it would be helpful.
> >
> > R1.7
> > > To further substantiate our claims that a variance-sensitive score function is a primary driver of excellent results on varsortable data, we have conceived an additional experiment. [...]
> >
> > Thank you for conducting this experiment. I am not sure whether I would personally, as a reviewer, recommend introducing the experiment into the _main_ paper because it was not part of the original review by me (and by other reviewers). Further questions/discussion might be required regarding this experiment.
> >
> > R1.8.
> > > Regarding societal impact, we disagree with a potential negative societal impact of our research. On the contrary, we believe it has a positive impact as it raises awareness of a previously overlooked benchmarking determinant and fosters a better understanding of their implicit assumptions relevant for real-world applications.
> >
> > I am sorry, just to clarify, I did mention in my original review that there are generally, potentially huge, possible impacts of this submission. Hence, I actually have agreed with you, to the best of my understanding, that the submission has (potentially) huge positive impacts
> >
> > However, I have thought that there is also a potential negative societal impact if e.g. there might be a potentially (inadvertent) contribution to e.g. temporary bias against approaches that rely on e.g. marginal variance ordering even though such ordering might be natural for many cases.
> >
> > It is likely a fair and valid point that positive societal impacts of this submission likely far outweighs possible negative ones, but I think it is important to note the latter, even though they might be significantly smaller.
> >
> > *
> >
> > If possible, could you comment regarding some of the points raised in section “Some other/minor points:” of my review (the ones that have not been addressed in your response), please?
> >
> > Looking forward to your response, if possible, as part of this review process.

---

> > > ### Author Response · Authors · 2021-08-18
> > > **Clarifying Explanations**
> > >
> > > Dear Reviewer,
> > >
> > > We are happy to hear that you found our reposponse helpful and that we could clarify your questions.
> > >
> > > ## Variance in ANMs and Known Identifiability Results
> > >
> > > We address the following two main items first, as we believe they are crucial for research on additive noise models and hope aligning on those will help clarify your points.
> > >
> > > ### Variance in ANMs
> > >
> > > > R1.1. “We obtain the variance of a node by adding each ancestors' noise variance multiplied with the squared product of the corresponding paths' coefficients to its own noise variance.”
> > > >
> > > > I am sorry, I might be missing something, but should potentially covariances of ancestors’ nodes be included there generally as well?
> > >
> > > Statement 3.2 in our first response is correct. A linear ANM is parameterized by edge weights and noise standard deviations (cf. Section 2.1 of our submission). Each node's variance can therefore be expressed in terms of those parameters. We recap this in the following exemplary 3-node SEM $A=N_A$, $B=w_{A\to B}A + N_B$, $C=w_{B\to C}B + N_C$, and noise variances $\sigma_A^2, \sigma_B^2, \sigma_C^2$. The marginal variances are then given as follows:
> > >
> > >   - $Var(A)=Var(N_A)=\sigma_A^2$.
> > >   - $Var(B)=Var(w_{A\to B}A + N_B)=Var(w_{A\to B}N_A + N_B)=w_{A->B}^2\sigma_A^2 + \sigma_B^2$
> > >   - $Var(C)=Var(w_{B->C}B+N_C)=w_{B->C}^2Var(B) + Var(N_C)=w_{B->C}^2(w_{A->B}^2\sigma_A^2 + \sigma_B^2)+\sigma_C^2$
> > >
> > > That is, the variance of each node is obtained by recursively following the structural equations and following the rules for the variance of linear combinations of independent random variables.
> > > Analogous calculations can be found in the linear ANM literature. For an example, see [Park 2020, Section 3], who focus on the corresponding conditional variance expressions to derive their identifiability results.
> > >
> > > ### Known Identifiability Results and Order-Parent Search Separation
> > >
> > > >>> Line 179: “If varsortability v = 1, the causal structure is identifiable” - this is a strong statement. I see how it could be potentially (sometimes?) identifiable (from a “partial” causal order), but I don’t see why the causal structure would be necessarily identifiable in general, especially from noisy data. A proof, a sketch and/or a reference would be very helpful.
> > > >>>
> > > >
> > > > Thank you very much. I think this clarification is helpful.
> > > >
> > > > However, while intuitively it makes sense that for some models under some assumptions and in limit we could recover structure, it still would be helpful to refer to/show a general proof of this. (For example, it is generally less intuitively obvious for cases with a structure where node A is a parent for B and C; and B is a parent for C as well.)
> > >
> > > Our statement relies on the following points:
> > > 1. Structure learning can be, and commonly is, separated into order search and parent selection (c.f. [Shimizu et al, 2011, (in particular Section 3.2)], [Chen et al, 2019], [Park, 2020, (in particular Algorithm 3)]).
> > > 2. In the case of varsortability v=1, identifiability of a causal order is immediate by virtue of the order of strictly increasing marginal variances agreeing with the causal order as per the definition of varsortability (c.f. Section 3.1).
> > > 3. Given a causal ordering, the causal structure can be inferred in the sample limit by variable-selection [Shojaie and Michailidis, 2010] and intervention distributions be estimated consistently [Bühlmann et al. 2014].
> > >
> > > To sum up, we rely on well-established identifiability results, all of which are referenced in our submission (c.f. Sections 2.2, 3.2), and the definition of varsortability (c.f. Section 3.1). Many settings of linear ANMs are already covered by existing identifiability results (c.f. Section 2.2). Our claim extends existing identifiability results only by the case of Gaussian ANMs that exhibit varsortability=1 but are not already covered by the identifiability results in Park, 2020 (see Appendix C).
> > >
> > > We hope this helps clarify the identifiability of different linear ANMs.
> > >
> > >
> > > > Could you advise me, please, when you refer to “Bühlmann et al. 2014”, do you mean “CAM: Causal additive models, high-dimensional order search and penalized regression”?
> > >
> > > Yes, in our article and response we refer to the following article by "Bühlmann et al. 2014":
> > > Bühlmann, J. Peters, J. Ernest. CAM: Causal Additive Models, high-dimensional Order Search and Penalized Regression, Annals of Statistics 42:2526-2556, 2014. DOI: 10.1214/14-AOS1260.
> > >
> > > Regarding your question, if varsortability=1, all causal ancestors of a node A come before A in the order of increasing marginal variance. Thus, sorting by marginal variance and filling in all edges from lower-variance nodes to higher-variance nodes yields a supergraph of the true causal graph. Variable selection techniques can then be used to prune this supergraph and remove any edges that are not present in the ground-truth graph.
> > >
> > > ## Further points
> > >
> > > ### R1.2 Marginal Variance _Tends_ To Increase
> > >
> > > Marginal variance does not necessarily increase along each path, but rather tends to increase as stated in the Abstract and throughout our submission. As you suggest, and as our title indicates, this is the case for existing synthetic data generation regimes commonly used for benchmarking.
> > > The increase of marginal variance along the causal order in common benchmarks can be seen by the performance of sortnregress. The effect is stark and may be surprising, precisely because marginal variance does not necessarily always increase. This is why we think it is important to raise awareness of this commonly overlooked or underestimated phenomenon in existing ANM simulation regimes and how it may dominate benchmarking performance.
> > > As we discuss in Section 6, future research into the phenomenon and its impact on benchmarks and real-world applicability of structure learning algorithms is needed and prompted for by our contribution.
> > > As stated in 3.2 of our original response, we will provide an additional illustration for a range of graph parameters in Appendix E.
> > >
> > > ### R1.3 Varsortability of 0.5 Is Not High
> > >
> > > A varsortability of 0.5 would mean that 50% of connected node pairs are correctly and 50% incorrectly ordered by marginal variance. This amounts to chance level in which case the performance of sortnregress degrades to that of randomregress (c.f. Appendix F.2). A value of 0.57 therefore is arguably not particularly high and substantially below what is observed in common benchmarks on raw data of linear ANMs with varsortabilities all exceeding 0.94 (cf. Table 4). We perform the real-world data analysis on 10 bootstrapped samples of the 853 observations to obtain confidence intervals.
> > >
> > > ### R1.5 Scale In Case of Identical Units
> > >
> > > For real-world data, data scales may be different even if the units are the same. For example, a shooter's distance from their target may affect their accuracy measured as distance between result and target. Although both are distances, it is not clear whether they should both be measured in, say, meters. This example also illustrates how in practice the effect variable's marginal variance may be smaller than that of the cause even if the same unit is used for both.
> > >
> > > ### R1.6 No Exact Match
> > >
> > > We do not claim an exact match and believe further research is needed to further delineate algorithm performance given the frequently overlapping boxes and inconsistent ordering for NOTEARS, GOLEM, and sortnregress in the settings considered in Section 4 and Appendix J (c.f. point 6 of our initial response).
> > >
> > > ### R1.8 Bias Against Varsortability-affected Algorithms
> > >
> > > We appreciate that the degree of varsortability in real-world data is an open question and it is not our intention to plainly discourage the use of varsortability-affected methods. Instead, our contribution aims to advance structure learning by highlighting a previously unknown driving factor of benchmarking performance. For this reason we use careful phrasing and focus on the severe performance difference between benchmarking on raw versus standardized data.
> > > To further reduce the risk of our contribution being misread, we will rephrase lines 375-377 as follows:
> > > _This begs the question what degree of varsortability can be observed or assumed in real-world data. If the marginal variances carry information about the causal order, our results suggest that it can and should be leveraged for structure learning. Otherwise, our contribution motivates future research into representative benchmarks and may put the practical applicability of the additive noise assumption into question._
> > >
> > > ### Other/Minor Points:
> > >
> > > In the interest of a focused discussion here, we address minor details such as rephrasings, typos, or definitions of acronyms, in the revised version directly. Thank you for your diligent reading!
> > >
> > > Regarding line 9, we refer to the case where varsortability is 0.5 and an ordering by increasing marginal variance is thus, on average, not more informative about the causal order than a random ordering. Thank you for pointing this out, we will clarify this in the revised version.
> > >
> > > Regarding line 43, we will reorder so that the definition appears earlier.
> > >
> > > We calculated the varsortability of the real-world dataset (see 3.3 of our original response).
> > >
> > >
> > > ### Summary
> > >
> > > In our responses, we have outlined how we will incorporate your suggestions and addressed clarifying questions in detail. Given that you appear to agree on the relevance, originality, and validity of our contribution, we hope this has eased your discomfort with accepting a paper that you describe as "probably [..] very beneficial for the field".

---

> > > > ### Comment · Reviewer_BrkD · 2021-08-23
> > > > **Response**
> > > >
> > > > Dear Authors,
> > > >
> > > > Thank you very much. Your responses are very helpful.
> > > >
> > > > I am sorry, I have two further comments:
> > > >
> > > > >> R1.1. “We obtain the variance of a node by adding each ancestors' noise variance multiplied with the squared product of the corresponding paths' coefficients to its own noise variance.”
> > > > >> I am sorry, I might be missing something, but should potentially covariances of ancestors’ nodes be included there generally as well?
> > > > > Statement 3.2 in our first response is correct.
> > > >
> > > > Thank you. I appreciate and it makes sense that generally if a model is parametrised by N parameters, then all properties of such model generally can be expressed as a function of those parameters (assuming those properties are expressible in general).
> > > >
> > > > Your example also makes sense. However, I still don't see why in linear ANMs covariance terms would be generally absent. For example, for the model below:
> > > > * $A = N_A$
> > > > * $B = w_1 A + N_B$
> > > > * $C = w_2 A + w_3 B + N_c$
> > > > to the best of my understanding, in the variance of $C$ there will be an extra term which is the two covariances between $w_2 A$ and $w_3 B$ (and that covariance is generally not equal to 0).
> > > >
> > > > Because of that, the statement "We obtain ..." does not make full sense to me. If you think I am mistaken, please, let me know.
> > > >
> > > > Moreover, I think your next suggested sentence "Cancellation of noise terms along different paths is possible but unlikely ..." looks consistent with the fact that there might be other term terms that are not just variances and squares of some values (which all would be positive).
> > > >
> > > > ***
> > > >
> > > > > Known Identifiability Results and Order-Parent Search Separation
> > > >
> > > > Thank you. In your last response you refer to [Shojaie and Michailidis, 2010] regarding causal structure inference:
> > > > > the causal structure can be inferred in the sample limit by variable-selection [Shojaie and Michailidis, 2010]
> > > > (and to [Bühlmann et al. 2014] regarding intervention distribution estimation)
> > > > but in your previous response you only referred to "Bühlmann et al. 2014".
> > > >
> > > > From my subjective point of view, it would be helpful to clarify in
> > > >
> > > > "This is the case because at v=1, an ordering by marginal variance is a valid causal ordering and sparse regression of each node onto its predecessors then enables reconstruction of the graph in the sample limit under mild assumptions (cf. Bühlmann et al. 2014)."
> > > >
> > > > what exact paper(s) do proof that (is it [Shojaie and Michailidis, 2010], [Bühlmann et al. 2014], both, (neither,) and under what specific assumptions (even if they are just listed in the reference paper(s))?).
> > > >
> > > > Your response to the part as below would be appreciated:
> > > > > If yes (I think so), and assuming you might be using version 2 (“1 Dec 2014”) from arXiv (arXiv:1310.1533), do you refer to section 2.5, page 11, equation (12), or/and to something else? If so, would that equation (as that paper writes: “[...] all relevant variables (i.e., edges) are selected”) cover the full statement about all cases where the varsortability equals 1 (e.g. including absences of edges in order to fully identify the causal structure)?
> > > >
> > > > ***
> > > >
> > > > Looking forward to your response, if possible, as part of this review process.

---

> > > > > ### Author Response · Authors · 2021-08-26
> > > > > **Marginal Variance in Linear ANMs and Identifiability Through Varsortability**
> > > > >
> > > > > Dear Reviewer,
> > > > >
> > > > > We are happy you found our response helpful. We address your two clarification questions below.
> > > > >
> > > > >
> > > > > ### Calculating Marginal Variances in Linear ANMs
> > > > >
> > > > > We understand that your question concerns our statement
> > > > >
> > > > > >>>> We obtain the variance of a node by adding each ancestors' noise variance multiplied with the squared product of the corresponding paths' coefficients to its own noise variance.
> > > > >
> > > > > For the exemplary linear ANM you mentioned, the marginal variances of A and B are calculated as in our previous example. For node C we have
> > > > >
> > > > > $Var(C) = Var(w_2 A + w_3 B + N_c)$ by following the structural equations
> > > > > $= Var(w_2 N_a + w_3 (w_1 A + N_b) + N_c)$ by following the structural equations
> > > > > $= Var(w_2 N_a + w_3 (w_1 N_a + N_b) + N_c)$ by following the structural equations
> > > > > $= Var((w_2+w_3w_1)N_a + w_3N_b + N_c)$ by rearranging
> > > > > $= (w_2+w_3w_1)^2 \sigma_a^2 + w_3^2 \sigma_b^2 + \sigma_c^2$ by rules for variance of linear combinations of random variables.
> > > > >
> > > > > Thus, each ancestor's (here, A and B) noise variance (here, $\sigma_a^2$ and $\sigma_b^2$) is multiplied by the squared product of the paths' coefficients and added to the nodes own noise variance (here, $\sigma_c^2$).
> > > > > It may not have been obvious how the weight products along different paths starting from the same nodes are aggregated. Your question prompted us to find a more procedural phrasing, which we will use in lines 174-176:
> > > > > _We obtain the marginal variance of a node by adding the variance contribution of all its ancestors to the node's own noise variance; to obtain the variance contribution of an ancestor, we take the product of the edge weights along each directed path from ancestor to node, sum these path coefficient products, square, and multiply with the ancestor's noise variance._
> > > > >
> > > > > It is, as you point out, equally possible to phrase this calculation in terms of variances and covariances:
> > > > > $Var(C) = Var(w_2 A) + Var(w_3 B) + 2Cov(w_2 A, w_3 B) + Var(N_c)$
> > > > > which simplifies to exactly the same expression as above via
> > > > > $= w_2^2 \sigma_a^2 + (w_1w_3)^2\sigma_a^2 + w_3^2 \sigma_b^2 + 2(w_2w_3 \cdot Cov(N_A,w_1N_A)) + \sigma_c^2$
> > > > > $= w_2^2 \sigma_a^2 + (w_1w_3)^2\sigma_a^2 + w_3^2 \sigma_b^2 + 2(w_1w_2w_3 \sigma_a^2) + \sigma_c^2$
> > > > > $= (w_2 + w_1w_3)^2 \sigma_a^2 + w_3^2 \sigma_b^2 + \sigma_c^2$.
> > > > > Thus agreeing with our expression above, which thence is correct and does not ignore any covariance terms.
> > > > > Further, we consider an expression in terms of model parameters (the one described above and stated in the last equation) more instructive because it allows us to draw a direct link between parameter sampling in benchmarks and varsortability.
> > > > >
> > > > > Exact cancellation only holds if $w_2+w_3w_1=0$, which results in a faithfulness violation (A and C would be independent, although connected in the causal graph). Such settings are unlikely under independent sampling of edge coefficients (cf. Meek 1995).
> > > > > If $w_2$ and $w_3w_1$ have opposing signs, partial cancellation of the noise contribution from $A$ is possible. At the same time, the magnitude of $w_3$ scales the variance contribution stemming from $N_b$, creating a subtle interplay. Empirically, despite the possibility for partial cancellation we observe a strong trend of increasing node variances along the causal order for the edge weight and noise variance distributions considered in common benchmarking settings.
> > > > >
> > > > > We hope this helps clarify the computation of marginal variances in linear ANMs.
> > > > >
> > > > >
> > > > > ### Identifiability
> > > > >
> > > > > > From my subjective point of view, it would be helpful to clarify in
> > > > > >
> > > > > > [...]
> > > > > >
> > > > > > what exact paper(s) do proof that (is it [Shojaie and Michailidis, 2010], [Bühlmann et al. 2014], both, (neither,) and under what specific assumptions (even if they are just listed in the reference paper(s))?).
> > > > >
> > > > > While the insight that an ordering by variance is a causal ordering under varsortability=1 is a contribution of ours, a self-contained exposition of existing results regarding identification of the causal structure given a causal order is beyond scope and page limit of our submission.
> > > > >
> > > > > To clarify and provide the relevant helpful pointers to the literature, as you suggest, we therefore propose to qualify the citation more explicitly as follows:
> > > > > _(see, for example, Section 2.5 of Bühlmann et al. 2014 for a discussion of parent selection and Shojaie and Michailidis, 2010 for the consistency of an adaptive lasso approach for edge selection given a valid causal order and under the usual regularity assumptions)._
> > > > >
> > > > >
> > > > > To put our presentation into context, please note that Shojaie and Michailidis, 2010 is entirely dedicated to the problem of edge selection given a causal order and Section 2.5 of Bühlmann et al. 2014 also addresses the question explicitly e.g. (page 2535): "The task of estimating such a super-DAG $\hat{D}^{\hat{π}} ⊇ D^0$ is conceptually straightforward: starting from the complete super-DAG $D^{\hat{π}}$ of $D^0$ as discussed above, we can use model selection or a penalized multivariate (auto-) regression technique in the model representation (5). For additive model fitting, we can either use hypothesis testing for additive models [15] or the Group Lasso [28], or its improved version with a sparsity-smoothness penalty proposed in [16]."
> > > > >
> > > > > We are not entirely sure we fully understand your last question about the coverage of Section 2.5. If varsortability=1, the causal order is known and the discussion of Section 2.5 in Bühlmann et al. 2014 for selecting the correct edges in an ANM given a causal order applies. As is common in hypothesis testing, the null is accepted (here meaning absence of an edge) unless the evidence is sufficient to reject the null (here concluding the presence of an edge). The mentioned screening property addresses the question whether all true edges are eventually selected by the Group Lasso. Shojaie and Michailidis, 2010 establish that "adaptive lasso can consistently estimate the true DAG under the usual regularity assumptions" (cf. Page 2 [Shojaie and Michailidis, 2010])."
> > > > >
> > > > > We hope the more detailed reference helps direct future readers to the relevant sections of the supporting literature.

---

> > > > > > ### Author Response · Authors · 2021-09-02
> > > > > > **Re:**
> > > > > >
> > > > > > Dear Reviewer,
> > > > > >
> > > > > > We thank you for engaging in a conversation with us and for assessing our work.
> > > > > >
> > > > > > Best regards,
> > > > > > The Authors

---

> > > > > > > ### Comment · Reviewer_BrkD · 2021-09-03
> > > > > > > **Response**
> > > > > > >
> > > > > > > Dear Authors,
> > > > > > >
> > > > > > > Thank you very much.
> > > > > > >
> > > > > > > I am sorry, two further comments (which should not affect my score):
> > > > > > >
> > > > > > > 1. Regarding the equation on the line after line 202 (in the main paper): it is derived in Appendix B. In Appendix B, if I understand correctly, it might be assumed that Var(A) is equal to Var(N_A) but I don't see why that would necessarily be the case (since A might have some ancestors in addition to its own noise). If possible, could you comment on that, please?
> > > > > > >
> > > > > > > 2. Following my original review regarding "More derivation details for section 3.4 might be helpful (e.g. in an appendix section).", I think it would be very helpful if further details are provided for section 3.4. I think that currently it is only a sketch and more details are required for that to be easily interpreted and comprehensively verified by a reader.
> > > > > > >
> > > > > > > I hope it helps.
> > > > > > >
> > > > > > > Yours faithfully,
> > > > > > > One of the reviewers

---

> > > > > > > > ### Author Response · Authors · 2021-09-03
> > > > > > > > **Re: further comments**
> > > > > > > >
> > > > > > > > Dear Reviewer,
> > > > > > > >
> > > > > > > > Thank you. We have fixed 1) to indicate the statement is intended for $A$ being a root node such that $Var(A)$ is equal to $Var(N_A)$. If $w \geq 1$, the pair is varsortable independent of whether A is a root node.
> > > > > > > > Thank you also for suggestion 2). Adding details of the derivation to the appendix is an excellent idea for better accessibility while allowing us to focus on our main topic of benchmarking within the page limit of the main text.
> > > > > > > > We will include a step-by-step derivation of the gradients without inline equations in Appendix B.
> > > > > > > >
> > > > > > > > Best regards,
> > > > > > > > The Authors

---

> > > > > > > > > ### Comment · Reviewer_BrkD · 2021-09-05
> > > > > > > > > **Response**
> > > > > > > > >
> > > > > > > > > Dear Authors,
> > > > > > > > >
> > > > > > > > > Thank you very much.
> > > > > > > > >
> > > > > > > > > > If ..., the pair is varsortable independent of whether A is a root node.
> > > > > > > > >
> > > > > > > > > Could you provide a proof, if possible, please?
> > > > > > > > >
> > > > > > > > > I am sorry, one more point (regarding one of the previous responses):
> > > > > > > > > > We will highlight that sensitivity to data scale is commented on in the references mentioned in line 53-56 ...
> > > > > > > > >
> > > > > > > > > I am sorry, just to double check, is that actually commented explicitly e.g. in (Zheng et al., 2018)?
> > > > > > > > >
> > > > > > > > > And a few more questions, if it is okay:
> > > > > > > > > 1. In section 2.1, on line 99, is there a chance that j=1,...,d should be k=1,...d?
> > > > > > > > > 2. Regarding section 3.4: generally a derivative of a logarithm does not have a logarithm (unless a logarithm is inside some function by itself). Is it expected that there are logarithms in gradients?
> > > > > > > > > 3. Is there a chance that e.g. the gradient of MSE(W) should be proportional to X^T (XW - X) (that is, should it be multiplied by -1)?
> > > > > > > > > 4. Is it assumed by any chance for section 3.4 that the observations have been adjusted to have zero means? (Otherwise, e.g., would necessarily high “MSE_j”-s (line 246) be reflecting high _variance_ nodes?)
> > > > > > > > > 5. Regarding “bigger gradient steps” (lines 241-242 and line 249) - are absolute values of gradient steps meant? (Since weights might be positive or negative.)
> > > > > > > > > 6. Are algorithms mentioned in section 3.4 usually initialised with the zero matrices?
> > > > > > > > > 7. You mention in the appendix (lines 804-805) that lambda was set to 0 for NOTEARS. Is there a chance that the results might have been different if lambda was not set to 0?
> > > > > > > > > 8. Is LogDet potentially missing from losses for GOLEM-s in the paper?
> > > > > > > > > 9. Regarding “empirical covariances” (line 237): are they “unnormalised” ones?
> > > > > > > > > 10. Regarding the same “empirical covariances”: is it assumed that residuals (in particular, after the first step) would have zero mean? Would it be the case in general?
> > > > > > > > > 11. Does “symmetric” e.g. on line 235 mean that w_{i,j} = w_{j,i}, or something else?
> > > > > > > > > 12. Regarding the “residuals” (from line 238): are they generally expected to be larger for the components with higher marginal variance? (It sounds like a reasonable guess but it is an interesting question whether it is actually the case in general.)
> > > > > > > > > 13. Even if the “residuals” (that relate to x_j) are larger for the components with higher marginal variances, given that there are parts of _something_ that reflect “covariances” with the other component being x_i, is it generally fair to assume that that _something_ will be larger for the components with higher marginal variances? (For example, might the contribution of x_i in the “covariances” have actually the opposite effect (i.e. regarding w_{i,j} vs w_{j,i})?)
> > > > > > > > > 14. “MSE” is used throughout the paper. Is it “mean” though if there is no division by “n”? Might it rather be e.g. LSL (as in “least squares loss”) or something like that?
> > > > > > > > >
> > > > > > > > > Yours faithfully,
> > > > > > > > > One of the reviewers

---

> > > > > > > > > > ### Author Response · Authors · 2021-09-07
> > > > > > > > > > **Re: Response**
> > > > > > > > > >
> > > > > > > > > > Dear Reviewer,
> > > > > > > > > >
> > > > > > > > > > We thank you for your continued interest.
> > > > > > > > > >
> > > > > > > > > > We are in fact unsure about the internal stage of the review process and whether reviewer assessments are concluded. Therefore, to minimize overhead for (S)ACs in handling our submission, we (propose to) focus on those aspects here that may be relevant for you to conclude (or update) your current assessment.
> > > > > > > > > > We are happy to answer your other follow-up questions further below.
> > > > > > > > > >
> > > > > > > > > > As evident from this exchange, our analysis of benchmarks on simulated ANM data and the concept of varsortability offer novel insights and stimulate future research into (continuous) structure learning, benchmarking procedures, and suitable assumptions for causal modelling of real world processes.
> > > > > > > > > > We expect that it will—similarly to this author-reviewer exchange—spur a fruitful discourse with the wider NeurIPS and structure learning community on how it should inform the current surge in the development of novel structure learning algorithms. We thus consider it a timely and valuable contribution.
> > > > > > > > > >
> > > > > > > > > > > > We will highlight that sensitivity to data scale is commented on in the references mentioned in line 53-56 ...
> > > > > > > > > > >
> > > > > > > > > > > I am sorry, just to double check, is that actually commented explicitly e.g. in (Zheng et al., 2018)?
> > > > > > > > > >
> > > > > > > > > > Zheng et al 2018 touches upon data scale implicitly in Appendix B and C. Data scale is often commented on in passing and briefly when discussing the plausibility of assumptions on the noise variance or weight scales. Our contribution stresses that data scale and the effect of standardization deserve more attention during the development and presentation of benchmark comparisons.
> > > > > > > > > >
> > > > > > > > > > > > If ..., the pair is varsortable independent of whether A is a root node.
> > > > > > > > > > >
> > > > > > > > > > > Could you provide a proof, if possible, please?
> > > > > > > > > >
> > > > > > > > > > For a variable pair $A \overset{w}{\to} B$ "without any common ancestors and no other directed path from A to B" and with $w \geq 1$, we have $Var(B) \geq w^2 Var(A) + \sigma_b^2 \geq Var(A)$. The first inequality is due to omission of any variance contribution to B from parents other than A (which are independent of A since there are no common ancestors to A and B and no further paths from A to B) and the second inequality is due to $w \geq 1$. This argument is independent of A being a root node.
> > > > > > > > > >
> > > > > > > > > > Best regards,
> > > > > > > > > > The Authors
> > > > > > > > > >
> > > > > > > > > >
> > > > > > > > > > ---
> > > > > > > > > >
> > > > > > > > > >
> > > > > > > > > > 1\. & 2. & 14. We have corrected the typos.
> > > > > > > > > >
> > > > > > > > > >
> > > > > > > > > > 3\. The sign is absorbed in the proportionality constant.
> > > > > > > > > >
> > > > > > > > > > 4\. & 10. As per model definition in Section 2.1 (cf. line 97), any linear combination of the variables in X (such as, for example, residuals), are in turn also a linear combination of the variables in N and thus have an expected value of 0. Therefore, for example, their variance (second central moment) is equal to the second moment and we do not need to estimate the expected value by taking the mean of the observations, but instead can subtract the known expected value (0) when estimating variances or covariances from observations.
> > > > > > > > > >
> > > > > > > > > > 5\. Yes.
> > > > > > > > > >
> > > > > > > > > > 6\. Yes.
> > > > > > > > > >
> > > > > > > > > > 7\. Hyperparameters of all methods (not only NOTEARS) may affect the quantitative results. We assessed a wide range of different hyperparameter settings with no changes to the qualitative results.
> > > > > > > > > > (For the specific case of NOTEARS, note that our choice of NOTEARS over the NOTEARS-L1 variant is in line with the findings of the original paper, in which regularization is proposed for small n, but did not lead to improvements for large n=1000 (cf. Zheng et al 2018, Sections 5.3 and Appendix D.3 and Tables 1 and 2 therein, and Figure 3a vs 3b).)
> > > > > > > > > >
> > > > > > > > > > 8\. No. Note that in line 142 we state the unnormalized (log-)likelihood-parts, the full loss function is stated in Appendix I.
> > > > > > > > > >
> > > > > > > > > > 9\. Yes.
> > > > > > > > > >
> > > > > > > > > > 11\. Yes, we are referring to symmetry of the (gradient) matrix.
> > > > > > > > > >
> > > > > > > > > > 12\. See our response to ciTV about lines 238/239.
> > > > > > > > > >
> > > > > > > > > > 13\. We agree that a conclusive explanation of the great/average performance of gradient-based methods on raw/standardized data is an interesting avenue for future research, especially as it may reveal ways to exploit/mitigate the effect of varsortability. Such an analysis is outside the scope of our paper. Section 3.4 complements our main focus on benchmarking and varsortability in simulated data by providing intuition and a starting point for future research (see first sentence of Section 3.4 "...how varsortability _may_ dominate...continuous structure learning...").

---

> > > > > > > > > > > ### Comment · Reviewer_BrkD · 2021-09-14
> > > > > > > > > > > **Response**
> > > > > > > > > > >
> > > > > > > > > > > Dear Authors,
> > > > > > > > > > >
> > > > > > > > > > > Thank you very much. It is very helpful.
> > > > > > > > > > >
> > > > > > > > > > > > Zheng et al 2018 touches upon data scale implicitly in Appendix B and C. Data scale is often commented on in passing and briefly when discussing the plausibility of assumptions on the noise variance or weight scales. We think that our contribution stresses that data scale and the effect of standardization deserve more attention during the development and presentation of benchmark comparisons.
> > > > > > > > > > >
> > > > > > > > > > > It is appreciated that data scales, weight scales and thresholding (as per (Zheng et al., 2018)) are related in some sense. However, I am still not sure whether e.g. the reference to (Zheng et al., 2018) on line 55 makes full sense as it is, I am sorry: as far as I understand, the explicit point about the data scale point (as per lines 53-56) regarding that method is part of the analysis of this particular submission (even though implicitly it might be related somehow), and it might be potentially confusing for a reader if it is found in “Introduction” (and not in “Introduction” / “Contribution”).
> > > > > > > > > > >
> > > > > > > > > > > > For a variable pair [...]
> > > > > > > > > > >
> > > > > > > > > > > Thank you. (That seems to hold indeed assuming the variance for B’s noise is not zero, which is assumed to be a general assumption.)
> > > > > > > > > > >
> > > > > > > > > > > ***
> > > > > > > > > > >
> > > > > > > > > > >
> > > > > > > > > > > Regarding 1, 2, 3, 4, 5, 6, 7, 8, 9, 10, 11, 14: thank you.
> > > > > > > > > > >
> > > > > > > > > > > Regarding 12: thank you. I think it would clarify things if it is discussed in the submission (including the info contained in the response to reviewer ciTV; in particular that the relevant part in the paper is based on empirical findings), if possible.
> > > > > > > > > > >
> > > > > > > > > > > >> 13. Even if the “residuals” (that relate to x_j) are larger for the components with higher marginal variances, given that there are parts of _something_ that reflect “covariances” with the other component being x_i, is it generally fair to assume that that _something_ will be larger for the components with higher marginal variances? (For example, might the contribution of x_i in the “covariances” have actually the opposite effect (i.e. regarding w_{i,j} vs w_{j,i})?)
> > > > > > > > > > >
> > > > > > > > > > > > We agree that a conclusive explanation of the great/average performance of gradient-based methods on raw/standardized data is an interesting avenue for future research, especially as it may reveal ways to exploit/mitigate the effect of varsortability. Such an analysis is outside the scope of our paper. Section 3.4 complements our main focus on benchmarking and varsortability in simulated data by providing intuition and a starting point for future research (see first sentence of Section 3.4 "...how varsortability may dominate...continuous structure learning...").
> > > > > > > > > > >
> > > > > > > > > > > Thank you. I am sorry, it is still not exactly clear why it is mentioned in the paper that “This leads to bigger gradient steps for edges pointing in the direction of high-variance nodes.” (lines 241-242). That is, the interplay between two parts of that unnormalised “covariance” is not obvious. Might I be missing something in that part of the analysis? Might it be based on some empirical findings (that are not mentioned in that section)? If possible, could you provide more details about this, please?
> > > > > > > > > > >
> > > > > > > > > > > Thank you,
> > > > > > > > > > > Yours faithfully,
> > > > > > > > > > > One of the reviewers

---

> > > > > > > > > > > > ### Author Response · Authors · 2021-09-21
> > > > > > > > > > > > **Causal Discovery Benchmarks May Be Easy to Game**
> > > > > > > > > > > >
> > > > > > > > > > > > Dear Reviewer,
> > > > > > > > > > > >
> > > > > > > > > > > > We appreciate your continued interest in the intricate role of varsortability for non-convex continuous optimization, and in particular for NOTEARS (Zheng et al., 2018).
> > > > > > > > > > > > We agree that this is an exciting topic which calls for further research but would like to note that the main contribution of our submission is to introduce the concept of varsortability, raise awareness of its presence in common benchmarks, and show that it can explain the remarkable results obtainable in common benchmarks by including the simple baseline *sortnregress*.
> > > > > > > > > > > > The concept of varsortability is novel and the impact of data scale / standardization on benchmarking performance has not been previously made explicit (see also reviewers ciTV & b4Yx).
> > > > > > > > > > > >
> > > > > > > > > > > > Best wishes,
> > > > > > > > > > > > The Authors.
> > > > > > > > > > > >
> > > > > > > > > > > >
> > > > > > > > > > > > ---
> > > > > > > > > > > >
> > > > > > > > > > > >
> > > > > > > > > > > > Besides the previously mentioned weight/threshold sensitivity experiments, Zheng et al. [2018] use the l2-loss and refer to its extensively studied statistical properties (Loh et al. [2014]).
> > > > > > > > > > > > One of these properties is scale-dependence (cf. Section 4.3 of Loh et al. [2014]).
> > > > > > > > > > > > In agreement with your original review, we rephrased 54-56 as follows to clarify that these articles are examples for which scale-sensitivity is known:
> > > > > > > > > > > >
> > > > > > > > > > > > > A researcher cannot rely on obtaining the same results for different measurement scales or after re-scaling the data when applying any method that leverages the data scale (examples include Peters and Bühlmann [2014], Park [2020], or Zheng et al [2018] who employ the least squares loss studied by Loh and Bühlmann [2014]).
> > > > > > > > > > > >
> > > > > > > > > > > > Re 12: We have added the information in our response to ciTV to the manuscript.
> > > > > > > > > > > >
> > > > > > > > > > > > Re 13: We agree that highlighting the empirical observation helps further clarify the heuristic nature of the argument (see also our exchange with reviewer ciTV). We rephrase _"As a result ... the causal direction."_ as follows:
> > > > > > > > > > > >
> > > > > > > > > > > > > In practice, we observe that during the next optimization steps $|\nabla MSE(W)|$ tends to be larger for edges pointing in the direction of nodes with high-variance residuals than for those pointing in the direction of nodes with low-variance residuals. Intuitively, when cycles are penalized, the insertion of edges pointing to nodes with high residuals is favoured as a larger reduction in MSE may be achieved than by including the opposing edge. Given high varsortability, this corresponds to favoring edges in the causal direction.
> > > > > > > > > > > >
> > > > > > > > > > > > Beyond that, we also observe that $W$ and its gradient depend on the implementation of the optimization routine. For example, the original implementation of NOTEARS fixes the diagonal of W at zero and leverages curvature information (L-BFGS-B), while GOLEM depends on learning rate optimizers.
> > > > > > > > > > > > We include this consideration in Appendix B (see our previous reply "Re: further comments") but consider a detailed analysis outside the scope of our submission, which is focused on benchmarking, sortnregress, and the concept/diagnostic tool of varsortability.
> > > > > > > > > > > > Future research is required to determine how precisely continuous structure learning algorithms achieve state-of-the-art results on highly varsortable data and, given our observations, we expect explanations to be specific to individual algorithms and their precise implementation.

---

> ### Comment · Reviewer_BrkD · 2021-09-02
> **Update**
>
> Dear Authors, Dear Other Reviewers, Dear Area Chairs, Dear All,
>
> Thanks for the other reviews and the authors' responses.
>
> The discussion with the authors has been very helpful indeed. Thanks a lot to the authors.
>
> Taking the discussion and the edits the authors have suggested (which I assume are going to be incorporated into a revised version if the submission is accepted) into account, I have changed my score to "6: Marginally above the acceptance threshold" (from "5: Marginally below the acceptance threshold").
>
> Regarding "a greedy forward DAG search (GDS) using a MSE score": my opinion is that it is probably the best not to include it into the main paper (i.e. for this conference if the submission is accepted) because the relevant parts have not been comprehensively reviewed; if it is nevertheless included in the main paper, I believe it must be mentioned that those parts have not been comprehensively peer-reviewed.
>
> Whilst I think the submission is very important and also significant, my opinion is that a higher score (i.e. higher than "6: Marginally above the acceptance threshold") can't be considered without a comprehensive review of a modified version.
>
> Yours faithfully & sincerely,
> One of the reviewers

---

### Author Response · Authors · 2021-09-02
**Summary of Changes for Camera-Ready Version**

Dear Reviewers and ACs,

We thank you for your time and feedback!

In our submission we reveal a previously unaddressed pattern in synthetic ANM data and its impact on causal discovery benchmarking.
The reviews appear to agree that this contribution would be valuable to the NeurIPS community.

Below we summarize the changes we incorporated for the camera-ready version.

1. We show the increase of marginal variance along the causal order for different graph parameters in appendix E (cf. BrkD).

2. We complement the existing experiments by including one additional method (GDS) to illustrate the role of scale-sensitive score functions in combinatorial methods (cf. ciTV), extend our preliminary nonlinear results (cf. b4Yx), and further contextualize the performances of NOTEARS, Golem, and sortnregress (cf. BrkD).
We concur with reviewer BrkD to detail this addition in the appendix and frame it as preliminary.

3. We incorporate all rephrasings/changes devised in correspondence with the reviewers to further enhance the clarity of our exposition and increase accessibility to a wider audience.


Best regards,
The Authors

---

### Decision · Program_Chairs · 2021-09-27

**Decision:**

Accept (Poster)

**Comment:**

A summary from one of the reviews:

"This paper shows that one needs to be cautious when rescaling your data prior to using a causal structure learning algorithms. The authors introduce the concept of varsortability that measures the agreement in how much the marginal variance tends to increase along the causal order. They show that standardization of your data can hurt performance in identifying the DAG (or its equivalence class) which can be explained by varsortability. The authors claim that this concept of varsortability also explains why even after standardization certain continuous structure learning algorithm perform well. The authors focuses on additive noise models and perform an extensive benchmark."

While the initial reviewer opinions were split, the eventual consensus on this paper is that it brings a valuable message of caution regarding developing and benchmarking causal inference methods with simulated data.